# Learning with Logical Constraints but without Shortcut Satisfaction

**Zenan Li**[1]**, Zehua Liu**[2]**, Yuan Yao**[1]**, Jingwei Xu**[1]**, Taolue Chen**[3]**, Xiaoxing Ma**[1]**, Jian Lü**[1]

[1]State Key Lab of Novel Software Technology, Nanjing University, China
[2]Department of Mathematics, The University of Hong Kong, Hong Kong
[3]Department of Computer Science, Birkbeck, University of London, UK
`lizn@smail.nju.edu.cn, liuzehua@connect.hku.hk,`
`t.chen@bbk.ac.uk, {y.yao,jingweix,xxm,lj}@nju.edu.cn`

## Abstract

Recent studies have explored the integration of logical knowledge into deep learning via encoding logical constraints as an additional loss function. However, existing approaches tend to vacuously satisfy logical constraints through shortcuts, failing to fully exploit the knowledge. In this paper, we present a new framework for learning with logical constraints. Specifically, we address the shortcut satisfaction issue by introducing dual variables for logical connectives, encoding how the constraint is satisfied. We further propose a variational framework where the encoded logical constraint is expressed as a distributional loss that is compatible with the model's original training loss. The theoretical analysis shows that the proposed approach bears salient properties, and the experimental evaluations demonstrate its superior performance in both model generalizability and constraint satisfaction.

## 1 Introduction

There have been renewed interests in equipping deep neural networks (DNNs) with symbolic knowledge such as logical constraints/formulas (Hu et al., 2016; Xu et al., 2018; Fischer et al., 2019; Nandwani et al., 2019; Li & Srikumar, 2019; Awasthi et al., 2020; Hoernle et al., 2021). Typically, existing work first translates the given logical constraint into a differentiable loss function, and then incorporates it as a penalty term in the original training loss of the DNN. The benefits of this integration have been well-demonstrated: it not only improves the performance, but also enhances the interpretability via regulating the model behavior to satisfy particular logical constraints.

Despite the encouraging progress, existing approaches tend to suffer from the *shortcut satisfaction problem*, i.e., the model overfits to a particular (easy) satisfying assignment of the given logical constraint. However, not all satisfying assignments are the truth, and different inputs may require different assignments to satisfy the same constraint. An illustrative example is given in Figure 1. Essentially, the example considers a logical constraint $P \rightarrow Q$, which holds when $(P, Q) = (\mathbf{T}, \mathbf{T})$ or $(P, Q) = (\mathbf{F}, \mathbf{F})/(\mathbf{F}, \mathbf{T})$. However, it is observed that existing approaches tend to simply satisfy the constraint via assigning $\mathbf{F}$ to $P$ for all inputs, even when the real meaning of the logic constraint is arguably $(P, Q) = (\mathbf{T}, \mathbf{T})$ for certain inputs (e.g., class '6' in the example).

To escape from the trap of shortcut satisfaction, we propose to consider *how* a logical constraint is satisfied by distinguishing between different satisfying assignments of the constraint for different inputs. The challenge here is the lack of direct supervision information of how a constraint is satisfied other than its truth value. However, our insight is that, by addressing this "harder" problem, we can make more room for the conciliation between logic information and training data, and achieve better model performance and logic satisfaction at the same time. To this end, when translating a logical constraint into a loss function, we introduce a *dual variable* for each operand of the logical connectives in the conjunctive normal form (CNF) of the logical constraint. The dual variables, together with the softened truth values for logical variables, provide a working interpretation for the satisfaction of the logical constraint. Take the example in Figure 1: for the satisfaction of $P \rightarrow Q$, we consider its CNF $\neg P \vee Q$ and introduce two variables $\tau_1$ and $\tau_2$ to indicate the weights of the

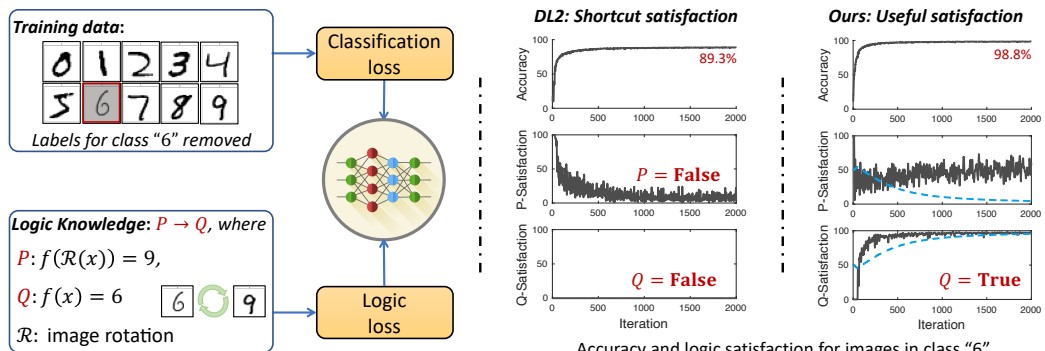

Figure 1: Consider a semi-supervised classification task of handwritten digit recognition. For the illustration purpose, we remove the labels of training images in class '6', but introduce a logical rule $P := (f(R(\mathbf{x})) = 9) \to Q := (f(\mathbf{x}) = 6)$ to predict '6', where $R(\mathbf{x})$ stands for rotating the image $\mathbf{x}$ by $180°$. The ideal satisfying assignments should be $(P, Q) = (\mathbf{T}, \mathbf{T})$ for class '6'. However, existing methods (e.g., DL2 (Fischer et al., 2019)) tend to vacuously satisfy the rule by discouraging the satisfaction of $P$ for all inputs, including those actually in class '6'. In contrast, our approach successfully learns to satisfy $Q$ when $P$ holds for class '6', even achieving comparable accuracy (98.8%) to the fully supervised setting.

softened truth values of $\neg P$ and $Q$, respectively. The blue dashed lines in the right part of Figure 1 (the $P$-Satisfaction and $Q$-Satisfaction subfigues) indicate that the dual variables gradually converge to the intended weights for class '6'.

Based on the dual variables, we then convert logical conjunction and disjunction into convex combinations of individual loss functions, which not only improves the training robustness, but also ensures *monotonicity* with respect to logical entailment, i.e., the smaller the loss, the higher the satisfaction. Note that most existing logic to loss translations do not enjoy this property but only ensure that the logical constraint is fully satisfied when the loss is zero; however, it is virtually infeasible to make the logical constraint fully satisfied in practice, rendering an unreliable training process towards constraint satisfaction.

Another limitation of existing approaches lies in the *incompatibility* during joint training. That is, existing work mainly treats the translated logic loss as a penalty under a multi-objective learning framework, whose effectiveness strongly relies on the weight selection of each objective, and may suffer when the objectives compete (Kendall et al., 2018; Sener & Koltun, 2018). In contrast, we introduce an additional random variable for the logical constraint to indicate its satisfaction degree, and formulate it as a distributional loss which is compatible with the neural network's original training loss under a variational framework. We cast the joint optimization of the prediction accuracy and constraint satisfaction as a game and propose a stochastic gradient descent ascent algorithm to solve it. Theoretical results show that the algorithm can successfully converge to a superset of local Nash equilibria, and thus settles the incompatibility problem to a large extent.

In summary, this paper makes the following main contributions: 1) a new logic encoding method that translates logical constraints to loss functions, considering how the constraints are satisfied, in particular, to avoid shortcut satisfaction; 2) a variational framework that jointly and compatibly trains both the translated logic loss and the original training loss with theoretically guaranteed convergence; 3) extensive empirical evaluations on various tasks demonstrating the superior performance in both accuracy and constraint satisfaction, confirming the efficacy of the proposed approach.

## 2 LOGIC TO LOSS FUNCTION TRANSLATION

### 2.1 LOGICAL CONSTRAINTS

For a given neural network, we denote the data point by $(\mathbf{x}, \mathbf{y}) \in \mathcal{X} \times \mathcal{Y}$, and use $\mathbf{w}$ to represent the model parameters. We use variable $v$ to denote the model's behavior of interest, which is represented

as a function $f_{\mathbf{w}}(\mathbf{x}, \mathbf{y})$ parameterized by $\mathbf{w}$. For instance, we can define $p_{\mathbf{w}}(y = 1 \mid \mathbf{x})$ as the (predictive) confidence of $y = 1$ given $\mathbf{x}$. We require $f_{\mathbf{w}}(\mathbf{x}, \mathbf{y})$ to be differentiable with respect to $\mathbf{w}$.

An *atomic formula* $a$ is in the form of $v \bowtie c$ where $\bowtie \in \{\leq, <, \geq, >, =, \neq\}$ and $c$ is a constant. We express a logical constraint in the form of a *logical formula*, consisting of usual conjunction, disjunction, and negation of atomic formulas. For instance, we can use $p_{\mathbf{w}}(y = 1 \mid \mathbf{x}) \geq 0.95 \vee p_{\mathbf{w}}(y = 1 \mid \mathbf{x}) \leq 0.05$ to specify that the confidence should be either no less than 95% or no greater than 5%. Note that $v = c$ can be written as $v \geq c \wedge v \leq c$, so henceforth for simplicity, we only have atomic formulas of the form $v \leq c$.

We use $\hat{v}$ to indicate a *state* of $v$, which is a concrete instantiation of $v$ over the current model parameters $\mathbf{w}$ and data point $(\mathbf{x}, \mathbf{y})$. We say a state $\hat{v}$ satisfies a logical formula $\alpha$, denoted by $\hat{v} \models \alpha$, if $\alpha$ holds under $\hat{v}$. For two logical formulas $\alpha$ and $\beta$, we say $\alpha \models \beta$ if any $\hat{v}$ that satisfies $\alpha$ also satisfies $\beta$. Moreover, we write $\alpha \equiv \beta$ if $\alpha$ is logically equivalent to $\beta$, i.e., they entail each other.

## 2.2 LOGICAL CONSTRAINT TRANSLATION

**(A) Atomic formulas**. For an atomic formula $a := v \leq c$, our goal is to learn the model parameters $\mathbf{w}^*$ to encourage that the current state $\hat{v}^* = f_{\mathbf{w}^*}(\mathbf{x}, \mathbf{y})$ can satisfy the formula for the given data point $(\mathbf{x}, \mathbf{y})$. For this purpose, we define a cost function

$$S_a(v) := \max(v - c, 0), \tag{1}$$

and switch to minimize it. It is not difficult to show that the atomic formula holds if and only if $\mathbf{w}^*$ is an optimal solution of $\min_{\mathbf{w}} S(\mathbf{w}) = \max(f_{\mathbf{w}}(\mathbf{x}, \mathbf{y}) - c, 0)$ with $S(\mathbf{w}^*) = 0$.

For example, for the predictive confidence $p_{\mathbf{w}}(y = 1 \mid \mathbf{x}) \geq 0.95$ introduced before, the corresponding cost function is defined as $\max(0.95 - p_{\mathbf{w}}(y = 1 \mid \mathbf{x}), 0)$, which is actually the norm distance between the current state $\hat{v}$ and the satisfied states of the atomic constraint. Such translation not only allows to find model parameters $\mathbf{w}^*$ efficiently by optimization, but also paves the way to encode more complex logical constraints as discussed in the following.

**(B) Logical conjunction**. For the logical conjunction $l := v_1 \leq c_1 \wedge v_2 \leq c_2$, where $v_i = f_{\mathbf{w}, i}(\mathbf{x}, \mathbf{y})$ for $i = 1, 2$, We first substitute the atomic formulas by their corresponding cost functions as defined in equation 1, and rewrite the conjunction as $(S(v_1) = 0) \wedge (S(v_2) = 0)$. Then, the conjunction can be equivalently converted into a maximization form, i.e., $\max(S(v_1), S(v_2)) = 0$. Based on this conversion, we may follow the existing work and define the cost function of logical conjunction as $S_{\wedge}(l) := \max(S(v_1), S(v_2))$. However, directly minimizing this cost function is less effective as it cannot well encode how the constraints are satisfied, and it is also not efficient since only one of $S(v_1)$ and $S(v_2)$ can be optimized in each iteration. Therefore, we introduce the dual variable $\tau_i$ for each atomic formula, and further extend it to the general conjunction case for $l := \wedge_{i=1}^k v_i \leq c_i$ as

$$S_{\wedge}(l) = \max_{\tau_1, \ldots, \tau_k} \sum_{i=1}^k \tau_i S(v_i), \quad \tau_1, \ldots, \tau_k \geq 0, \quad \tau_1 + \cdots + \tau_k = 1. \tag{2}$$

**(C) Logical disjunction**. Similar to conjunction, the logical disjunction $l := \vee_{i=1}^k v_i \leq c_i$. can be equivalently encoded into a minimization form. Hence, the corresponding cost function is as follows,

$$S_{\vee}(l) = \min_{\tau_1, \ldots, \tau_k} \sum_{i=1}^k \tau_i S(v_i), \quad \tau_1, \ldots, \tau_k \geq 0, \quad \tau_1 + \cdots + \tau_k = 1. \tag{3}$$

**(D) Clausal Formulas**. In general, for logical formula $\alpha = \wedge_{i \in \mathcal{I}} \vee_{j \in \mathcal{J}} v_{ij} \leq c_{ij}$ in the conjunctive normal form (CNF), the corresponding cost function can be defined as

$$S_{\alpha}(v) = \max_{\mu_i} \min_{\nu_{ij}} \sum_{i \in \mathcal{I}} \sum_{j \in \mathcal{J}^{(i)}} \mu_i \cdot \nu_{ij} \cdot \max(v_{ij} - c_{ij}, 0)$$

$$\text{s.t.} \sum_{i \in \mathcal{I}} \mu_i = 1, \sum_{j \in \mathcal{J}^{(i)}} \nu_{ij} = 1, \ \mu_i, \nu_{ij} \geq 0, \tag{4}$$

where $\mu_i (i \in \mathcal{I})$ and $\nu_{ij} (j \in \mathcal{J}^{(i)})$ are the dual variables for conjunction and disjunction, respectively.

The proposed cost function establishes an equivalence between the logical formula and the optimization problem. This is summarized in the following theorem, whose proof is included in Appendix A.

**Theorem 1.** *Given the logical formula $\alpha = \wedge_{i \in \mathcal{I}} \vee_{j \in \mathcal{J}} v_{ij} \leq c_{ij}$, if the dual variables $\{\mu_i, i \in \mathcal{I}\}$ and $\{\nu_{ij}, j \in \mathcal{J}\}$ of $S_\alpha(v)$ converge to $\{\mu_i^*, i \in \mathcal{I}\}$ and $\{\nu_{ij}^*, j \in \mathcal{J}\}$, then the cost function of $\alpha$ can be computed as $S_\alpha(v) = \max_{i \in \mathcal{I}} \min_{j \in \mathcal{J}} \{S_{ij} := \max(v_{ij} - c_{ij}, 0)\}$. Furthermore, the sufficient and necessary condition of $\hat{v}^* = f_{\mathbf{w}^*}(\mathbf{x}, \mathbf{y}) \models \alpha$ is that $\mathbf{w}^*$ is the optimal solution of $\min_{\mathbf{w}} S_\alpha(\mathbf{w})$, with the optimal value $S_\alpha(\mathbf{w}^*) = 0$.*

## 2.3 Advantages of Our Translation

**Monotonicity**. As shown in Theorem 1, the optimal $\mathbf{w}^*$ ensures that model's behavior $\hat{v}^* = f_{\mathbf{w}^*}(\mathbf{x}, \mathbf{y})$ can satisfy the target logical constraint $\alpha$. Unfortunately, $\mathbf{w}^*$ usually cannot achieve the optimum due to the coupling between the cost function $S_\alpha(v)$ and the original training loss as well as the fact that $S_\alpha(v)$ is usually non-convex. This essentially reveals the main rationale of our logical encoding, and we conclude it in the following theorem with the proof given in Appendix B.

**Theorem 2.** *For two logical formulas $\alpha = \wedge_{i \in \mathcal{I}} \vee_{j \in \mathcal{J}} a_{ij}^{(\alpha)}$ and $\beta = \wedge_{i \in \mathcal{I}} \vee_{j \in \mathcal{J}} a_{ij}^{(\beta)}$, we have $\alpha \models \beta$ if and only if $S_\alpha(v) \geq S_\beta(v)$ holds for any state of $v$.*

*Remarks*. Theorem 2 essentially states that, when dual variables converge, we can progressively achieve a higher satisfaction degree of the logical constraints, as the cost function continues to decrease. This is especially important in practice since it is usually infeasible to make the logical constraint fully satisfied.

**Interpretability**. The introduced dual variables control the satisfaction degree of each individual atomic formula, and gradually converge towards the best valuation. Elaborately, the dual variables essentially learn how the given logical constraint can be satisfied for each data point, which resembles the structural parameter used in Bengio et al. (2020) to disentangle causal mechanisms. Namely, the optimal dual variables $\tau^*$ disentangle the satisfaction degree of the entire logical formula into individual atomic formulas, and reveal the contribution of each atomic constraint to the entire logical constraint in a discriminative way. Consider the cost function $S_l(v) = \max_{\tau \in [0,1]} \tau S(v_1) + (1 - \tau)S(v_2)$ for $l := a_1 \vee a_2$. The expectation $E_v[\tau^*]$ estimates the probability of $p(a_1 \to l \mid \mathbf{x})$, and a larger $E_v[\tau^*]$ indicates a greater probability that the first atomic constraint is met.

**Robustness improvement**. The dual variables also improve the stability of numerical computations. First, compared with some classic fuzzy logic operators (e.g., directly adopting max and min operators to replace the conjunction and disjunction (Zadeh, 1965; Elkan et al., 1994; Hájek, 2013)), the dual variables alleviate the sensitivity to the initial point. Second, compared with other commonly-used translation strategies (e.g., using addition and multiplication in DL2 (Fischer et al., 2019)), the dual variables balance the magnitude of the cost function, and further avoid some bad stationary points. Some concrete examples are included in Appendix C.

# 3 A Variational Learning Framework

## 3.1 Distributional Loss for Logical Constraints

Existing work mainly adopts a multi-objective learning framework to integrate logical constraints into DNNs, which is sensitive to the weight selection of each individual loss. In this work, we propose a variational framework (Blei et al., 2017) to achieve better compatability between the two loss functions without the need of weight selection. Generally speaking, the original training loss for a neural network is usually a distributional loss (e.g., cross entropy), which aims to construct a parametric distribution $p_{\mathbf{w}}(\mathbf{y}|\mathbf{x})$ that is close to the target distribution $p(\mathbf{y}|\mathbf{x})$. To keep the compatibility, we extend the distributional loss to the logical constraints. More concretely, we define an additional $m$-dimensional random variable $\mathbf{z}$ for the target logical constraint $\alpha$ and define $\mathbf{z} = S_\alpha(\mathbf{v})$, where $\mathbf{v} = f_{\mathbf{w}}(\mathbf{x}, \mathbf{y})$. Here, we use vector $\mathbf{z}$ to indicate the combination of multiple variables (e.g., $v_1, \cdots, v_m$) in the logical constraint for training efficiency. [1]

---

[1] Note that $\mathbf{z}$ can be considered as a random variable even when the data point $(\mathbf{x}, \mathbf{y})$ is given because there usually exists randomness when training the model (e.g., dropout, batch-normalization, etc.).

Next, we frame the training as the distributional approximation between the parametric distribution and target distribution over $\mathbf{y}$ and $\mathbf{z}$, and choose the KL divergence as the distributional loss,

$$\min_{\mathbf{w}} \sum_{i=1}^{N} \mathrm{KL}(p(\mathbf{y}_i, \mathbf{z}_i | \mathbf{x}_i) \| p_{\mathbf{w}}(\mathbf{y}_i, \mathbf{z}_i | \mathbf{x}_i)). \tag{5}$$

By using Bayes' theorem, $p(\mathbf{y}, \mathbf{z}|\mathbf{x})$ can be decomposed as $p(\mathbf{y}, \mathbf{z}|\mathbf{x}) = p(\mathbf{z}|(\mathbf{x}, \mathbf{y})) \cdot p(\mathbf{y}|\mathbf{x})$. Therefore, we can reformulate equation 5 as:

$$\min_{\mathbf{w}} \sum_{i=1}^{N} \mathrm{KL}(p(\mathbf{y}_i|\mathbf{x}_i) \| p_{\mathbf{w}}(\mathbf{y}_i|\mathbf{x}_i)) + \mathrm{E}_{\mathbf{y}_i|\mathbf{x}_i}[\mathrm{KL}(p(\mathbf{z}_i|\mathbf{x}_i, \mathbf{y}_i) \| p_{\mathbf{w}}(\mathbf{z}_i|\mathbf{x}_i, \mathbf{y}_i))]. \tag{6}$$

The detailed derivation can be found in Appendix D. In equation 6, the first term is the original training loss, and the second term is the distributional loss of the logical constraint.

The remaining question is how to model the conditional probability distribution of the random variable $\mathbf{z}$. Note that $\mathbf{z} = 0$ indicates that the target logical constraint $\alpha$ is satisfied. Thus, the target distribution of $\mathbf{z}$ can be defined as a Dirac delta distribution (Dirac, 1981, Sec. 15), i.e., $p(\mathbf{z}|\mathbf{x}, \mathbf{y})$ is a zero-centered Gaussian with variance tending to zero (Morse & Feshbach, 1954, Chap. 4.8). Furthermore, considering the non-negativity of $\mathbf{z}$, we form the Dirac delta distribution as the limit of the sequence of truncated normal distributions on $[0, +\infty)$:

$$p(\mathbf{z}|\mathbf{x}, \mathbf{y}) = \lim_{\sigma \to 0} \mathcal{TN}(\mathbf{z}; 0, \sigma^2 \mathbf{I}) = \lim_{\sigma \to 0} (\frac{2}{\sigma})^m \phi(\frac{\mathbf{z}}{\sigma}),$$

where $\phi(\cdot)$ is the probability density function of the standard multivariate normal distribution. For the parametric distribution of $\mathbf{z}$, we model $p_{\mathbf{w}}(\mathbf{z}|\mathbf{x}, \mathbf{y})$ as the truncated normal distribution on $[0, +\infty)$ with mean $\boldsymbol{\mu} = S_\alpha(\mathbf{w})$ and covariance $\mathrm{diag}(\boldsymbol{\delta}^2)$,

$$p_{\mathbf{w}}(\mathbf{z}|\mathbf{x}, \mathbf{y}) = \frac{1}{\sqrt{|\mathrm{diag}(\boldsymbol{\delta}^2)|}} \frac{\phi(\frac{\mathbf{z}-\boldsymbol{\mu}}{\boldsymbol{\delta}})}{1 - \Phi(-\frac{\boldsymbol{\mu}}{\boldsymbol{\delta}})},$$

where $\Phi(\cdot)$ is the cumulative distribution function of the standard multivariate normal distribution, and $\frac{\boldsymbol{\mu}}{\boldsymbol{\delta}}$ denotes element-wise division of vectors $\boldsymbol{\mu}$ and $\boldsymbol{\delta}$. Hence, the final optimization problem of our learning framework is

$$\min_{\mathbf{w}, \boldsymbol{\delta}} \sum_{i=1}^{N} \mathrm{KL}(p(\mathbf{y}_i|\mathbf{x}_i) \| p_{\mathbf{w}}(\mathbf{y}_i|\mathbf{x}_i)) + \log|\mathrm{diag}(\boldsymbol{\delta})| + \frac{1}{2}\|\frac{\boldsymbol{\mu}_i}{\boldsymbol{\delta}}\|^2 + \log(1 - \Phi(-\frac{\boldsymbol{\mu}_i}{\boldsymbol{\delta}})), \tag{7}$$

where $\boldsymbol{\mu}_i = S_\alpha(\mathbf{v}_i), \mathbf{v}_i = f_{\mathbf{w}}(\mathbf{x}_i, \mathbf{y}_i)$. Different from the probabilistic soft logic that estimates the probability for each atomic formula, we establish the probability distribution $\mathcal{TN}(\mathbf{z}; \boldsymbol{\mu}, \mathrm{diag}(\boldsymbol{\delta}^2))$ of the target logical constraint. This allows to directly determine the variance $\boldsymbol{\delta}^2$ of the entire logical constraint rather than analyzing the correlation between atomic constraints.

Moreover, the minimizations of $\mathbf{w}$ and $\boldsymbol{\delta}$ can be viewed as a competitive game (Roughgarden, 2010; Schaefer & Anandkumar, 2019). Roughly speaking, they first cooperate to achieve both higher model accuracy and higher degree of logical constraint satisfaction, and then compete between these two sides to finally reach a (local) Nash equilibrium (even though it may not exist). The local Nash equilibrium means that the model accuracy and logical constraint satisfaction cannot be improved at the same time, which shows a compatibility in the sense of game theory.

### 3.2 Optimization Procedure

Note that the dual variables $\boldsymbol{\tau}_\wedge$ and $\boldsymbol{\tau}_\vee$ of the cost function $\boldsymbol{\mu}_i = S_\alpha(\mathbf{v}_i)$ also need to be optimized during training. To solve the optimization problem, we utilize a stochastic gradient descent ascent (SGDA) algorithm (with min-oracle) (Lin et al., 2020), and the details of the training algorithm is summarized in Appendix F. Specifically, we update $\mathbf{w}$ and $\boldsymbol{\tau}_\vee$ through gradient descent, update $\boldsymbol{\tau}_\wedge$ through gradient ascent, and update $\boldsymbol{\delta}$ by its approximate min-oracle, i.e., via its upper bound $\mathrm{KL}(p(\mathbf{z}|\mathbf{x}, \mathbf{y}) \| p_{\mathbf{w}}(\mathbf{z}|\mathbf{x}, \mathbf{y})) \leq \log|\mathrm{diag}(\boldsymbol{\delta})| + \frac{1}{2}\|\frac{\boldsymbol{\mu}}{\boldsymbol{\delta}}\|^2 + \mathrm{const}$. Thus, the update of $\boldsymbol{\delta}$ in each iteration is $\boldsymbol{\delta}^2 = \frac{1}{N}\sum_{i=1}^{N} \boldsymbol{\mu}_i = \frac{1}{N}\sum_{i=1}^{N} S_\alpha(\mathbf{v}_i)$.

The update of $\delta$ plays an important role in our algorithm, because the classic SGDA algorithm (i.e., direct alternating gradient descent on $\mathbf{w}$ and $\delta$) may converge to limit cycles. Therefore, updating $\delta$ via its approximate min-oracle not only makes the iteration more efficient, but also ensures the convergence of our algorithm as summarized in Theorem 3.

**Theorem 3.** *Let* $\upsilon(\cdot) = \min_\delta L(\cdot, \delta)$*, and assume* $L(\cdot)$ *is* $L$-Lipschitz. *Algorithm 1 with step size* $\eta_\mathbf{w} = \gamma/\sqrt{T+1}$ *ensures the output* $\overline{\mathbf{w}}$ *of* $T$ *iterations satisfies*

$$\mathbb{E}[\|\nabla e_{\upsilon/2\kappa}(\overline{\mathbf{w}})\|^2] \leq O\left(\frac{\kappa\gamma^2 L^2 + \Delta_0}{\gamma\sqrt{T+1}}\right),$$

*where* $e_\upsilon(\cdot)$ *is the Moreau envelop of* $\upsilon(\cdot)$*.*

*Remarks.* Theorem 3 states that the proposed algorithm with suitable stepsize can successfully converge to a point $(\mathbf{w}^*, \delta^*, \tau_\wedge^*, \tau_\vee^*)$, where $(\mathbf{w}^*, \delta^*)$ is an approximate stationary point,[2] and $(\tau_\wedge^*, \tau_\vee^*)$ is a global minimax point. More detailed analysis and proofs are provided in Appendix G.

## 4 EXPERIMENTS AND RESULTS

We carry out experiments on four tasks, i.e., handwritten digit recognition, handwritten formula recognition, shortest distance prediction in a weighted graph, and CIFAR100 image classification. For each task, we train the model with normal cross-entropy loss on the labeled data as the baseline result, and compare our approach with PD (Nandwani et al., 2019) and DL2 (Fischer et al., 2019), which are the state-of-the-art approaches that incorporate logical constraints into the trained models. For PD, there are two choices (Choice 1 and Choice 2 named by the authors) to translate logical constraints into loss functions which are denoted by $PD_1$ and $PD_2$, respectively. We also compare with SL (Xu et al., 2018) and DPL (Manhaeve et al., 2018) in the first task. These two methods are intractable in the other three tasks as they both employ the knowledge compilation (Darwiche & Marquis, 2002) to translate logical constraint, which involves an implicit enumeration of all satisfying assignments. Each reported experimental result is derived by computing the average of five repeats. For each atomic formula $a := v \leq c$, we consider it is satisfied if $\hat{v} \leq c - \text{tol}$ where tol is a predefined tolerance threshold to relax the strict inequalities. We set $\text{tol} = 0.01$ on the three classification tasks and $\text{tol} = 1$ on the regression task (shortest distance prediction), respectively. More setup details can be found in Appendix H. The code, together with the experimental data, is available at https://github.com/SoftWiser-group/NeSy-without-Shortcuts.

### 4.1 HANDWRITTEN DIGIT RECOGNITION

In the first experiment, we construct a semi-supervised classification task by removing the labels of '6' in the MNIST dataset (LeCun et al., 1989) during training. We then apply a logical rule to predict label '6' using the rotation relation between '6' and '9' as $f(\tilde{\mathbf{x}}) = 9 \rightarrow f(\mathbf{x}) = 6$, where $\mathbf{x}$ is the input, and $\tilde{\mathbf{x}}$ denotes the result of rotating $\mathbf{x}$ by 180 degrees. We rewrite the above rule as the disjunction $(f(\tilde{\mathbf{x}}) \neq 9) \vee (f(\mathbf{x}) = 6)$. We train the LeNet model on the MNIST dataset, and further validate the transferability performance of the model on the USPS dataset (Hull, 1994).

The classification accuracy and logical constraint satisfaction results of class '6' are shown in Table 1. The ¬P-Sat. and Q-Sat. in the table indicate the satisfaction degrees of $f(\tilde{\mathbf{x}}) \neq 9$ and $f(\mathbf{x}) = 6$, respectively. At first glance, it is a bit strange that our approach significantly outperforms some alternatives on the accuracy, but achieves almost the same logic rule satisfaction. However, taking a closer look at how the logic rule is satisfied, we find that alternatives are all prone to learn an *shortcut* satisfaction (i.e., $f(\tilde{\mathbf{x}}) \neq 9$) for the logical constraint, and thus cannot further improve the accuracy. In contrast, our approach learns to satisfy $f(\mathbf{x}) = 6$ when $f(R(\mathbf{x})) = 9$, which is supposed to be learned from the logical rule.

On the USPS dataset, we additionally train a reference model with full labels, and it achieves 82.1% accuracy. Observe that the domain shift deceases the accuracy of all methods, but our approach still obtains the highest accuracy (even comparable with the reference model) and constraint satisfaction. This is due to the fact that our approach better learns the prior knowledge in the logical constraint and

---

[2]The set of stationary points contains local Nash equilibria (Jin et al., 2020). In other words, all local Nash equilibria are stationary points, but not vice versa.

Table 1: Results (%) of the handwritten digit recognition task. The proposed approach learns how the logical constraint is satisfied (i.e., the Q-Sat.) while the existing methods fail.

| Method | MNIST | | | | USPS | | | |
|---|---|---|---|---|---|---|---|---|
| | Acc. | Sat. | ¬P-Sat. | Q-Sat. | Acc. | Sat. | ¬P-Sat. | Q-Sat. |
| Baseline | 89.4 | 27.4 | 27.4 | 0.0 | 75.4 | 81.1 | 81.1 | 0.0 |
| SL | 85.5 | 97.0 | 97.0 | 0.0 | 48.7 | 98.8 | 98.8 | 0.0 |
| DPL | 88.0 | 78.0 | 78.0 | 0.0 | 25.6 | 8.8 | 8.8 | 0.0 |
| $PD_1$ | 89.2 | 44.2 | 44.2 | 0.0 | 73.7 | 87.0 | 87.0 | 0.0 |
| $PD_2$ | 89.3 | 63.1 | 63.1 | 0.0 | 74.3 | 87.6 | 87.6 | 0.0 |
| DL2 | 89.3 | 91.3 | 91.3 | 0.0 | 69.9 | 98.8 | 98.8 | 0.0 |
| Ours | **98.8** | **98.9** | 49.7 | **97.7** | **80.2** | **98.8** | 90.5 | **75.2** |

Table 2: Results (%) of the handwritten formula recognition task. The proposed approach achieves the best results.

| Method | 2/80* Split | | 5/20* Split | |
|---|---|---|---|---|
| | Acc. | Sat. | Acc. | Sat. |
| Baseline | 57.4 | 64.0 | 75.6 | 81.5 |
| $PD_1$ | 62.1 | 63.2 | 76.5 | 78.6 |
| $PD_2$ | 62.5 | 62.5 | 76.4 | 78.4 |
| DL2 | 62.9 | 57.5 | 79.3 | 86.6 |
| Ours | **65.1** | **82.4** | **79.5** | **92.8** |

Table 3: Results on the shortest distance prediction task. The proposed approach achieves the best results.

| Method | MSE / MAE | Sat. (%) |
|---|---|---|
| Baseline | 8.90 / 2.07 | 43.9 |
| $PD_2$ | 9.11 / 2.11 | 46.1 |
| DL2 | 14.21 / 2.68 | 65.3 |
| Ours | **7.09 / 1.87** | **69.0** |

\* It refers to the proportion of labeled and unlabeled data used from the training set.

thus yields better transferability. An additional experiment of a transfer learning task on the CIFAR10 dataset showing the transferability of our approach can be found in Appendix J.

## 4.2 HANDWRITTEN FORMULA RECOGNITION

We next evaluate our approach on a handwritten formula (HWF) dataset, which consists of 10K training formulas and 2K test formulas (Li et al., 2020). The task is to predict the formula from the raw image, and then calculate the final result based on the prediction. We adopt a simple grammar rule in the mathematical formula, i.e., four basic operators $(+, -, \times, \div)$ cannot appear in adjacent positions. For example, "$3 + \times 9$" is not a valid formula according to the grammar. Hence, given a formula with $k$ symbols, the target logical constraint is formulated as $l := \wedge_{i=1}^{k-1}(a_{i1} \vee a_{i2})$, with $a_{i1} := p_{\text{digits}}(\mathbf{x}_i) + p_{\text{digits}}(\mathbf{x}_{i+1}) = 2$ and $a_{i2} := p_{\text{ops}}(\mathbf{x}_i) + p_{\text{ops}}(\mathbf{x}_{i+1}) = 1$, where $p_{\text{digits}}(\mathbf{x})$ and $p_{\text{ops}}(\mathbf{x})$ represent the predictive probability that $\mathbf{x}$ is a digit or an operator, respectively. This logical constraint emphasizes that either any two adjacent symbols are both numbers (encoded by $a_{i1}$), or only one of them is a basic operator (encoded by $a_{i2}$).

To show the efficacy of our approach, we cast this task in a semi-supervised setting, in which only a very small part of labeled data is available, but the logical constraints of unlabeled data can be used during the training process. The calculation accuracy and logical constraint satisfaction results are provided in Table 2. In the table, we consider two settings of data splits, i.e., using 2% labeled data and 80% unlabeled data, as well as using 5% labeled data and 20% unlabeled data in the training set. It is observed that our approach achieves the highest accuracy and logical constraint satisfaction in both cases, demonstrating the effectiveness of the proposed approach.

## 4.3 SHORTEST DISTANCE PREDICTION

We next consider a regression task, i.e., to predict the length of the shortest path between two vertices in a weighted connected graph $\mathcal{G} = (V, E)$. Basic properties should be respected by the model's prediction $f(\cdot)$, i.e., for any $v_i, v_j, v_k \in V$, the prediction should obey (1) symmetry:

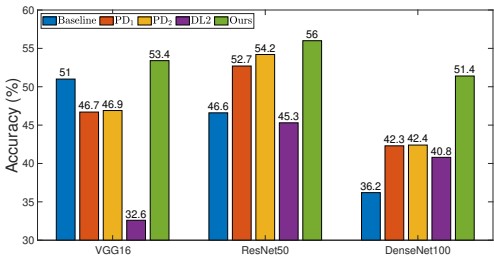

(a) Accuracy of class classification

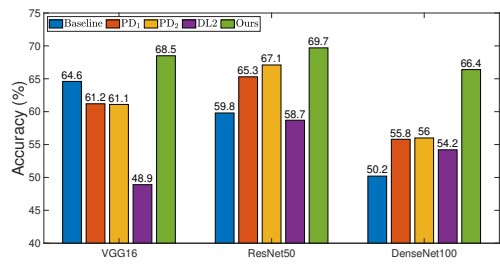

(b) Accuracy of superclass classification

Figure 2: The accuracy results (%) of image classification on the CIFAR100 dataset. The proposed approach outperforms the competitors in all the three cases for both class and superclass classification.

$f(v_i, v_j) = f(v_j, v_i)$ and (2) triangle inequality: $f(v_i, v_j) \leq f(v_i, v_k) + f(v_k, v_j)$. We train an MLP that takes the adjacency matrix of the graph as the input, and outputs the predicted shortest distances from the source node to all the other nodes. In the experiment, the number of vertices is fixed to $15$, and the weights of edges are uniformly sampled among $\{1, 2, \ldots, 9\}$.

The results are shown in Table 3. We do not include the results of $PD_1$ as it does not support this regression task ($PD_1$ only supports the case when the softened truth value belongs to $[0, 1]$). It is observed that our approach achieves the lowest MSE/MAE with the highest constraint satisfaction. We further investigated the reason of the performance gain, and found that models trained by the existing methods easily overfit to the training set (e.g., training MSE/MAE nearly vanished). In contrast, the model trained by our method is well regulated by the required logical constraints, making the test error more consistent with the training error.

## 4.4 IMAGE CLASSIFICATION

We finally evaluate our method on the classic CIFAR100 image classification task. The CIFAR100 dataset contains 100 classes which can be further grouped into 20 superclasses (Krizhevsky et al., 2009). Hence, we use the logical constraint proposed by Fischer et al. (2019), i.e., if the model classifies the image into any class, the predictive probability of corresponding superclasses should first achieve full mass. For example, the *people* superclass consists of five classes (*baby*, *boy*, *girl*, *man*, and *woman*), and thus $p_{\text{people}}(\mathbf{x})$ should have $100\%$ probability if the input $\mathbf{x}$ is classified as girl. We can formulate this logical constraint as $\bigwedge_{s \in \text{superclasses}} (p_s(\mathbf{x}) = 0.0\% \lor p_s(\mathbf{x}) = 100.0\%)$, where the probability of the superclass is the sum of its corresponding classes' probabilities (for example, $p_{\text{people}}(\mathbf{x}) = p_{\text{baby}}(\mathbf{x}) + p_{\text{boy}}(\mathbf{x}) + p_{\text{girl}}(\mathbf{x}) + p_{\text{man}}(\mathbf{x}) + p_{\text{woman}}(\mathbf{x}))$.

We construct a labeled dataset of 10,000 examples and an unlabeled dataset of 30,000 examples by randomly sampling from the original training data, and then train three different models (VGG16, ResNet50, and DenseNet100) to evaluate the performance of different methods. The results of classification accuracy and constraint satisfaction are shown in Figure 2 and Table 8, respectively. We can observe significant enhancements in both metrics of our approach.

## 5 RELATED WORK

**Learning with constraints.** Research on linking learning and logical reasoning has emerged for years (Roth & Yih, 2004; Chang et al., 2008; Cropper & Dumančić, 2020). Early research mostly concentrated on mining well-generalized logical relations (Muggleton, 1992), i.e., inducing a set of logical (implication) rules based on given logical atoms and examples. Recently, several work switch to training the model with deterministic logical constraints that are explicitly provided (Kimmig et al., 2012; Giannini et al., 2019; Nandwani et al., 2019; van Krieken et al., 2022; Giunchiglia et al., 2022). They usually use an interpreter to soften the truth value, and utilize fuzzy logic (Wierman, 2016) to encode the logical connectives. However, such conversion is usually quite costly, as they necessitate the use of an interpreter, which should be additionally constructed by the most probable explanation (Bach et al., 2017). Moreover, the precise logical meaning is lost due to the introduction of

Table 4: The constraint satisfaction results (%) of image classification on the CIFAR100 dataset. The proposed approach outperforms the competitors in all the three cases.

| Method | VGG16 | ResNet50 | DenseNet100 |
|---|---|---|---|
| Baseline | 63.0 | 48.5 | 39.7 |
| $PD_1$ | 57.7 | 63.9 | 53.5 |
| $PD_2$ | 57.5 | 69.2 | 54.9 |
| DL2 | 31.2 | 78.4 | 62.2 |
| Ours | **81.1** | **87.2** | **88.2** |

the interpreter and it is also unclear if such encoding bears important properties such as monotonicity. To address this issue, Fischer et al. (2019) abandon the interpreter, and use addition/multiplication to encode logical conjunction/disjunction; Yang et al. (2022) apply straight-through estimators to encode CNF into a loss function. However, both of them still lack the monotonicity property. Xu et al. (2018) propose a semantic loss to impose Boolean constraints on the output layer to ensure the monotonicity property. However, its encoding rule does not discriminate different satisfying assignments, causing the shortcut satisfaction problem. Hoernle et al. (2021) aim at training a neural network that fully satisfies logical constraints, and hence directly restrict the model's output to the constraint. Although they introduce a variable to choose which constraint should be satisfied, the variable is categorical and thus limited to mutually exclusive disjunction.

**Neuron-symbolic learning.** Our work is related to neuro-symbolic computation (Garcez et al., 2019; Marra et al., 2021). In this area, integrating learning into logic has been explored in several directions including neural theorem proving (Rocktäschel & Riedel, 2017; Minervini et al., 2020), extending logic programs with neural predicates (Manhaeve et al., 2018; Yang et al., 2020), encoding algorithmic layers (e.g., satisfiability solver) into DNNs (Wang et al., 2019; Chen et al., 2020), using neural models to approach executable logical programs (Li & Srikumar, 2019; Badreddine et al., 2022; Ahmed et al., 2022), as well as incorporating neural networks into logic reasoning (Yang et al., 2017; Evans & Grefenstette, 2018; Dong et al., 2019). Along this direction, there are other ways to integrate logical knowledge into learning, e.g., knowledge distillation (Hu et al., 2016), learning embeddings for logical rules (Xie et al., 2019), treating rules as noisy labels (Awasthi et al., 2020), and abductive reasoning (Dai et al., 2019; Zhou, 2019).

**Multi-objective learning.** Training model with logical constraints is essentially a multi-objective learning task, where two typical solutions exist (Marler & Arora, 2004), i.e., the $\epsilon$-*constraint* method and the *weighted sum* method. The former rewrites objective functions into constraints, i.e., solving $\min_x f(x)$, s.t., $g(x) \leq \epsilon$ instead of the original problem $\min_x(f(x), g(x))$. E.g., Donti et al. (2021) directly solve the learning problem with (hard) logical constraints via constraint completion and correction. However, this method may not be sufficiently efficient for deep learning tasks considering the high computational cost of Hessian-vector computation and the ill-conditionedness of the problem. The latter method is relatively more popular, which minimizes a proxy objective, i.e., a weighted average $\min_x w_1 f(x) + w_2 g(x)$. Such method strongly depends on the weights of the two terms, and may be highly ineffective when the two losses conflict (Kendall et al., 2018; Sener & Koltun, 2018).

## 6 CONCLUSION

In this paper, we have presented a new approach for better integrating logical constraints into deep neural networks. The proposed approach encodes logical constraints into a distributional loss that is compatible with the original training loss, guaranteeing monotonicity for logical entailment, significantly improving the interpretability and robustness, and avoiding shortcut satisfaction of the logical constraints at large. The proposed approach has been shown to be able to improve both model generalizability and logical constraint satisfaction. A limitation of the work is that we set the target distribution of any logical formula as the Dirac distribution, but further investigation is needed to decide when such setting could be effective and whether an alternative could be better. Additionally, our approach relies on the quality of the manually inputted logical formulas, and complementing it with automatic logic induction from raw data is an interesting future direction.

ACKNOWLEDGMENT

We are thankful to the anonymous reviewers for their helpful comments. This work is supported by the National Natural Science Foundation of China (Grants #62025202, #62172199). T. Chen is also partially supported by Birkbeck BEI School Project (EFFECT) and an overseas grant of the State Key Laboratory of Novel Software Technology under Grant #KFKT2022A03. Yuan Yao (y.yao@nju.edu.cn) and Xiaoxing Ma (xxm@nju.edu.cn) are the corresponding authors.

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

# A  PROOF OF THEOREM 1

## A.1  TRANSLATION EQUIVALENCE OF LOGICAL CONJUNCTION

The following proposition shows the equivalence of our translation and the original expression of logical conjunction.

**Proposition 1.** *Given $l := \bigwedge_{i=1}^{k} v_i \leq c_i$ where $v_i = f_{\mathbf{w},i}(\mathbf{x}, \mathbf{y})$ and $S_i := \max(v_i - c_i, 0)$ for $i = 1, \ldots, k$, if $\{\tau_i^*\}_{i=1}^{k}$ are the optimal solution of the cost function $S_\wedge(l)$,*

$$\sum_{i=1}^{k} \tau_i^* S(v_i) = \max(S(v_1), \ldots, S(v_k)).$$

*Furthermore, $\hat{v}^* = f_{\mathbf{w}^*}(\mathbf{x}, \mathbf{y}) \models l$ if and only if $\mathbf{w}^*$ is the optimal solution of $\min_{\mathbf{w}} S_\wedge(\mathbf{w})$, with the optimal value*

$$S_\wedge(\mathbf{w}^*) = \max(S_1(\mathbf{w}^*), \ldots, S_k(\mathbf{w}^*)) = 0.$$

*Proof.* For the minimization of the cost function $S_\wedge(\mathbf{w})$

$$\min_{\mathbf{w}} S_\wedge(\mathbf{w}) := \max(S_1(\mathbf{w}), \ldots, S_k(\mathbf{w})), \tag{8}$$

we introduce a slack variable $t$, and rewrite Eq. equation 8 as

$$\min_{\mathbf{w},t} t, \quad \text{s.t.,} \ t \geq S_i(\mathbf{w}), \ i = 1, \ldots, k. \tag{9}$$

The Lagrangian function of Eq. equation 9 is

$$L(\mathbf{w}, t; \tau_1, \ldots, \tau_k) = t + \sum_{i=1}^{k} \tau_i(S_i(\mathbf{w}) - t),$$

where $\tau_i \geq 0, i = 1, \ldots, k$. Let the gradient of $t$ vanish, we can obtain the dual problem:

$$\max_{\tau_1, \ldots, \tau_k} \min_{\mathbf{w}} \tau_1 S_1(\mathbf{w}) + \cdots + \tau_k S_k(\mathbf{w}),$$
$$\text{s.t.} \quad \tau_1 + \cdots + \tau_k = 1,$$
$$0 \leq \tau_i \leq 1, i = 1, \ldots, k.$$

By using the max-min inequality, we have

$$\max_{\tau_1, \ldots, \tau_k} \min_{\mathbf{w}} \tau_1 S_1(\mathbf{w}) + \cdots + \tau_k S_k(\mathbf{w}) \leq \min_{\mathbf{w}} \max_{\tau_1, \ldots, \tau_k} \tau_1 S_1(\mathbf{w}) + \cdots + \tau_k S_k(\mathbf{w}).$$

Therefore, the cost function $S_\wedge(\mathbf{w})$ of the logical conjunction can be computed by introducing the dual variables $\tau_i, i = 1, \ldots, k$:

$$S_\wedge(\mathbf{w}) = \max_{\substack{\tau_1, \ldots, \tau_k \in [0,1] \\ \tau_1 + \cdots + \tau_k = 1}} \tau_1 S_1(\mathbf{w}) + \cdots + \tau_k S_k(\mathbf{w}).$$

The Karush Kuhn–Tucker (KKT) condition of Eq. equation 9 is

$$\tau_1 \nabla S_1(\mathbf{w}) + \cdots + \tau_k \nabla S_k(\mathbf{w}) = 0,$$
$$\tau_1 + \ldots, \tau_k = 1,$$
$$t \geq S_i(\mathbf{w}), \tau_i \geq 0, \quad i = 1, \ldots, k,$$
$$\tau_i(t - S_i(\mathbf{w})) = 0, \quad i = 1, \ldots, k.$$

We denote the index set of the largest element in the set $\{S_i\}_{i=1}^{k}$ by $\mathcal{I}$. Suppose dual variables $\{\tau_i\}_{i=1}^{k}$ converge to $\{\tau_i^*\}_{i=1}^{k}$, then $\tau_j^* = 0$ for any $j \notin \mathcal{I}$, and thus $\sum_{i \in \mathcal{I}} \tau_i^* = 1$. Since $S_i(\mathbf{w}) = \max(S_1(\mathbf{w}), \ldots, S_k(\mathbf{w}))$ for any $i \in \mathcal{I}$, we have

$$S_\wedge(\mathbf{w}) = \tau_1^* S_1(\mathbf{w}) + \cdots + \tau_k^* S_k(\mathbf{w})$$
$$= \sum_{i \in \mathcal{I}} \tau_i^* S_i(\mathbf{w}) = \max(S_1(\mathbf{w}), \ldots, S_k(\mathbf{w})).$$

Furthermore, if $\mathbf{w}^*$ is the optimal solution and $S_\wedge(\mathbf{w}^*) = 0$, we have

$$S_1(\mathbf{w}^*) = \cdots = S_k(\mathbf{w}^*) = 0,$$

which implies that $\mathbf{w}^*$ ensures the satisfaction of the constraint $\bigwedge_{i=1}^k v_i \leq c_i$.

On the other hand, if $\mathbf{v}^* = f_{\mathbf{w}^*}(\mathbf{x}, \mathbf{y})$ entails the logical conjunction $\bigwedge_{i=1}^k v_i \leq c_i$, then we have $S_i(\mathbf{w}^*) = 0$ for $i = 1, \ldots, k$. Therefore, we have $\mathbf{w}^*$ is the optimal solution of $S_\wedge(\mathbf{w}^*)$ with $S_\wedge(\mathbf{w}^*) = 0$. □

## A.2 TRANSLATION EQUIVALENCE OF LOGICAL DISJUNCTION

We also have the following proposition demonstrating the equivalence between the cost function $S_\vee(l)$ and the original expression of logical disjunction.

**Proposition 2.** *Given $l := \bigvee_{i=1}^k v_i \leq c_i$, where $v_i = f_{\mathbf{w},i}(\mathbf{x}, \mathbf{y})$ and $S_i := \max(v_i - c_i, 0)$ for $i = 1, \ldots, k$, if $\{\tau_i^*\}_{i=1}^k$ are the optimal solution of the cost function $S_\vee(l)$, we have*

$$\sum_{i=1}^k \tau_i^* S(v_i) = \min(S(v_1), \ldots, S(v_k)).$$

*Furthermore, $\hat{v}^* = f_{\mathbf{w}^*}(\mathbf{x}, \mathbf{y}) \models l$ if and only if $\mathbf{w}^*$ is the optimal solution of $\min_{\mathbf{w}} S_\vee(\mathbf{w})$, with the optimal value*

$$S_\vee(\mathbf{w}^*) = \min(S_1(\mathbf{w}^*), \ldots, S_k(\mathbf{w}^*)) = 0.$$

*Proof.* For the minimization of the cost function $S_\vee(\mathbf{w})$

$$\min_{\mathbf{w}} S_\vee(\mathbf{w}) := \min(S_1(\mathbf{w}), \ldots, S_k(\mathbf{w})), \tag{10}$$

we introduce a slack variable $t$, and rewrite Eq. equation 10 as

$$\min_{\mathbf{w}} \max_t t, \quad \text{s.t.}, \ t \leq S_i(\mathbf{w}), \ i = 1, \ldots, k. \tag{11}$$

The corresponding dual problem is

$$\min_{\tau_1, \ldots, \tau_k} \min_{\mathbf{w}} \tau_1 S_1(\mathbf{w}) + \cdots + \tau_k S_k(\mathbf{w}),$$
$$\text{s.t.} \quad \tau_1 + \cdots + \tau_k = 1,$$
$$0 \leq \tau_i \leq 1, i = 1, \ldots, k.$$

Therefore, the cost function $S_\vee(\mathbf{w})$ of the logical disjunction can be computed by introducing the dual variables $\tau_i, i = 1, \ldots, k$:

$$S_\vee(\mathbf{w}) = \min_{\substack{\tau_1, \ldots, \tau_k \in [0,1] \\ \tau_1 + \cdots + \tau_k = 1}} \tau_1 S_1(\mathbf{w}) + \cdots + \tau_k S_k(\mathbf{w}).$$

Similar to the proof of Proposition 1, by using the KKT condition of Eq. equation 11, and supposing dual variables $\{\tau_i\}_{i=1}^k$ converge to $\{\tau_i^*\}_{i=1}^k$, we can obtain that

$$S_\vee(\mathbf{w}) = \tau_1^* S_1(\mathbf{w}) + \cdots + \tau_k^* S_k(\mathbf{w})$$
$$= \sum_{i \in \mathcal{I}} \tau_i^* S_i(\mathbf{w}) = \min(S_1(\mathbf{w}), \ldots, S_k(\mathbf{w})).$$

If $\mathbf{w}^*$ is the optimal solution and $S_\vee(\mathbf{w}^*) = 0$, there exists $S_i(\mathbf{w}^*) = 0$, which implies that $\mathbf{w}^*$ ensures the satisfaction of the constraint $\bigvee_{i=1}^k v_i \leq c_i$.

On the other hand, if $\mathbf{w}^*$ entails the logical disjunction $\bigvee_{i=1}^k v_i \leq c_i$, then there exists $S_i(\mathbf{w}^*) = 0$. Therefore, we have $\mathbf{w}^*$ is the optimal solution of $S_\vee(\mathbf{w}^*)$ with $S_\vee(\mathbf{w}^*) = 0$. □

## A.3 PROOF OF THEOREM 1

The proof can be directly derived from Proposition 1 and Proposition 2.

## B    PROOF OF THEOREM 2

*Proof.* We first denote all variables $v_1, \ldots, v_k$ involved in the logical constraint by a vector $\boldsymbol{v}$, and in this sense the corresponding constants $c_i, i = 1, \ldots, k$, in the logical constraint should be extended to $\mathbb{R} \cup \{+\infty\}$ such that the constraints $\alpha$ and $\beta$ can be written as

$$\alpha := \wedge_{i \in \mathcal{I}} \vee_{j \in \mathcal{J}} (\boldsymbol{v} \leq \boldsymbol{c}_{ij}^{(\alpha)}), \quad \beta := \wedge_{i \in \mathcal{I}} \vee_{j \in \mathcal{J}} (\boldsymbol{v} \leq \boldsymbol{c}_{ij}^{(\beta)}).$$

For the sufficient condition, suppose that $\boldsymbol{v}^* \models \alpha$. Then, we have $S_\alpha(\boldsymbol{v}^*) = 0$ by using Theorem 1. Since $S_\alpha(\boldsymbol{v}) \geq S_\beta(\boldsymbol{v})$ holds for any $\boldsymbol{v} \in \mathcal{V}$, and the cost function is non-negative, we can obtain that $S_\beta(\boldsymbol{v}^*) = 0$, which implies that $v^* \models \beta$.

For the necessary condition of Theorem 2, we introduce the following proposition.

**Proposition 3.** *Given the underlying space $\mathcal{V}$, for any point $\boldsymbol{v} \in \mathcal{V}$ and subset $C \subseteq \mathcal{V}$, we define the distance of $\boldsymbol{v}$ from $C$ as*

$$\mathrm{dist}(\boldsymbol{v}, C) = \inf\{\mathrm{dist}(\boldsymbol{v}, \boldsymbol{u}) \mid \boldsymbol{u} \in C\}.$$

*Moreover, let $A$ and $B$ be two closed subsets of $\mathcal{V}$, if $A$ is not an empty set, then the sufficient and necessary condition for $A \subseteq B$ is that*

$$\mathrm{dist}(\boldsymbol{v}, A) \geq \mathrm{dist}(\boldsymbol{v}, B), \quad \forall \boldsymbol{v} \in \mathcal{V}.$$

Therefore, let $A$ and $B$ be two sets implied by $\alpha$ and $\beta$, i.e.,

$$A = \cap_{i \in \mathcal{I}} \cup_{j \in \mathcal{J}} \{\boldsymbol{v} \mid \boldsymbol{v} \leq \boldsymbol{c}_{ij}^{(\alpha)}\}, \quad B = \cap_{i \in \mathcal{I}} \cup_{j \in \mathcal{J}} \{\boldsymbol{v} \mid \boldsymbol{v} \leq \boldsymbol{c}_{ij}^{(\beta)}\},$$

by using Proposition 3, we can obtain that $\alpha \models \beta$ if and only if $\mathrm{dist}(\boldsymbol{v}, A) \geq \mathrm{dist}(\boldsymbol{v}, B)$ for any $v \in \mathcal{V}$.

Given atomic formulas $a_1 := \boldsymbol{v} \leq \boldsymbol{c}_1$ and $a_2 := \boldsymbol{v} \leq \boldsymbol{c}_2$, the corresponding cost functions are $S_1(\boldsymbol{v}) = \max(\boldsymbol{v} - \boldsymbol{c}_1, 0)$ and $S_2(\boldsymbol{v}) = \max(\boldsymbol{v} - \boldsymbol{c}_2, 0)$, respectively. Then, the cost functions are essentially the (Chebyshev) distances of $\boldsymbol{v}$ to the constraints $\{\boldsymbol{v} \mid \boldsymbol{v} \leq \boldsymbol{c}_1\}$ and $\{\boldsymbol{v} \mid \boldsymbol{v} \leq c_2\}$, respectively. (It should be noted that $\boldsymbol{v} \leq \boldsymbol{c}$ can be decomposed into the conjunction of $v_i \leq c_i, i = 1, \ldots, k$). Hence, through Proposition 3, the sufficient and necessary condition of $a_1 \models a_2$ is that $S_{a_1}(\boldsymbol{v}) \geq S_{a_2}(\boldsymbol{v})$ for any $\boldsymbol{v} \in \mathcal{V}$.

Now, the rest is to prove that the cost functions of logical conjunction and disjunction are still the distance of $\boldsymbol{v}$ from the corresponding constraint, and it is not difficult to derive the results through a few calculations of linear algebra. One can also obtain a direct result by using Martinón (2004, Theorem 1). □

## C    CONCRETE EXAMPLES IN ROBUSTNESS IMPROVEMENT

(1) Let us consider a logical constraint $(v^2 \leq -1) \vee (3v \geq 2)$, the corresponding cost function based on the min and max operators is $S(v) = \min(v^2 + 1, \max(2 - 3v, 0))$. If we directly minimize $S(v)$ and set the initial point by $v_0 = 0$, we will have $S(v_0) = v_0^2 + 1 = 1$ and $\nabla_{\boldsymbol{\pi}} S(v_0) = 2v_0 = 0$. Hence, $v_0$ has already been an optimal solution of $\min S(v)$. In this case, the conventional optimization technique is not effectual, and cannot find a feasible solution that entails the disjunction even though it exists. Nevertheless, with the dual variable $\tau$, the minimization problem is $\min_{v,\tau} \tau(v^2 + 1) + (1 - \tau)(\max(2 - v, 0))$. Given initial points $v_0 = 0$ and $\tau_0 = 0.5$, one can easily obtain a feasible solution $v^* = 1.5$ via the coordinate descent algorithm.

(2) For translation strategy used in DL2, the cost functions of conjunction $a_1 \wedge a_2$ and disjunction $a_1 \vee a_2$ are defined by $S_\wedge(v) = S(v_1) + S(v_2)$ and $S_\vee(v) = S(v_1) \cdot S(v_2)$, respectively. The conjunction translation is essentially a special case of our encoding strategy (i.e., $\tau_1$ and $\tau_2$ are fixed to 0.5). For the disjunction, the multiplication may ruin the magnitude of the cost function, making it no longer a reasonable measure of constraint satisfaction. Moreover, this translation method also brings more difficulties to numerical calculations. For example, considering the disjunction constraint $(v = 1) \vee (v = 2) \vee (v = 3)$, $S_\vee(v) = S(v_1) \cdot S(v_2) \cdot S(v_3)$ induces two more bad stationary points (i.e., $v = 1.5$ and $v = 2.5$) compared with $\min_{v,\tau} S_\vee(v) = \tau_1 S(v_1) + \tau_2 S(v_2) + \tau_3 S(v_3)$.

## D    THE COMPUTATION OF KL DIVERGENCE

For input-output pair $(\mathbf{x}, \mathbf{y})$, with the logical variable $\mathbf{z}$, the KL divergence is

$$\mathrm{KL}(p(\mathbf{y}, \mathbf{z} \mid \mathbf{x}) \mid p_{\mathbf{w}}(\mathbf{y}, \mathbf{z} \mid \mathbf{x}))$$

$$= \int_{\mathbf{y}, \mathbf{z}} p(\mathbf{y}, \mathbf{z} \mid \mathbf{x}) \log \frac{p(\mathbf{y}, \mathbf{z} \mid \mathbf{x})}{p_{\mathbf{w}}(\mathbf{y}, \mathbf{z} \mid \mathbf{x})}$$

$$= \int_{\mathbf{y}, \mathbf{z}} p(\mathbf{y}, \mathbf{z} \mid \mathbf{x}) \left( \log \frac{p(\mathbf{y} \mid \mathbf{x})}{p_{\mathbf{w}}(\mathbf{y} \mid \mathbf{x})} + \log \frac{p(\mathbf{z} \mid \mathbf{x}, \mathbf{y})}{p_{\mathbf{w}}(\mathbf{z} \mid \mathbf{x}, \mathbf{y})} \right)$$

For the first term in RHS,

$$\int_{\mathbf{y}, \mathbf{z}} p(\mathbf{y}, \mathbf{z} \mid \mathbf{x}) \log \frac{p(\mathbf{y} \mid \mathbf{x})}{p_{\mathbf{w}}(\mathbf{y} \mid \mathbf{x})} = \mathrm{KL}(p(\mathbf{y} \mid \mathbf{x}) \| p_{\mathbf{w}}(\mathbf{y} \mid \mathbf{x})),$$

and for the second term in RHS,

$$\int_{\mathbf{y}, \mathbf{z}} p(\mathbf{y}, \mathbf{z} \mid \mathbf{x}) \log \frac{p(\mathbf{z} \mid \mathbf{x}, \mathbf{y})}{p_{\mathbf{w}}(\mathbf{z} \mid \mathbf{x}, \mathbf{y})} = \int_{\mathbf{y}, \mathbf{z}} p(\mathbf{y} \mid \mathbf{x}) p(\mathbf{z} \mid \mathbf{x}, \mathbf{y}) \log \frac{p(\mathbf{z} \mid \mathbf{x}, \mathbf{y})}{p_{\mathbf{w}}(\mathbf{z} \mid \mathbf{x}, \mathbf{y})}$$

$$= \mathbb{E}_{\mathbf{y} \mid \mathbf{x}} \left[ \int_{\mathbf{z}} p(\mathbf{z} \mid \mathbf{x}, \mathbf{y}) \log \frac{p(\mathbf{z} \mid \mathbf{x}, \mathbf{y})}{p_{\mathbf{w}}(\mathbf{z} \mid \mathbf{x}, \mathbf{y})} \right]$$

$$= \mathbb{E}_{\mathbf{y} \mid \mathbf{x}} [\mathrm{KL}(p(\mathbf{z} \mid \mathbf{x}, \mathbf{y}) \| p_{\mathbf{w}}(\mathbf{z} \mid \mathbf{x}, \mathbf{y}))].$$

It follows that

$$\mathrm{KL}(p(\mathbf{y}, \mathbf{z} \mid \mathbf{x}) \| p_{\mathbf{w}}(\mathbf{y}, \mathbf{z} \mid \mathbf{x})) = \mathrm{KL}(p(\mathbf{y} \mid \mathbf{x}) \| p_{\mathbf{w}}(\mathbf{y} \mid \mathbf{x})) + \mathbb{E}_{\mathbf{y} \mid \mathbf{x}} [\mathrm{KL}(p(\mathbf{z} \mid \mathbf{x}, \mathbf{y}) \| p_{\mathbf{w}}(\mathbf{z} \mid \mathbf{x}, \mathbf{y}))].$$

## E    KL DIVERGENCE OF TRUNCATED GAUSSIANS

Given two normal distributions truncated on $[0, +\infty)$ with means $\mu_1$ and $\mu_2$ and variances $\sigma_1^2$ and $\sigma_2^2$, the KL divergence can be computed as (Choudrey, 2002)[Appendix A.5]

$$\mathrm{KL}(\mathcal{TN}_1 \| \mathcal{TN}_2) = \frac{1}{2} \left[ \left( \frac{\sigma_1^2}{\sigma_2^2} - 1 \right) - \log \frac{\sigma_1^2}{\sigma_2^2} + \frac{(\mu_1 - \mu_2)^2}{\sigma_2^2} \right]$$

$$+ [(\frac{1}{\sigma_1^2} + \frac{1}{\sigma_2^2}) \mu_1 - \frac{2\mu_2}{\sigma_2^2}] \frac{\sigma_1}{\sqrt{2\pi}} \frac{1}{\exp(\frac{\mu_1^2}{2\sigma_1^2})(1 - \mathrm{erf}(-\frac{\mu_1}{\sqrt{2}\sigma_1}))}$$

$$+ \log \left( \frac{1 - \mathrm{erf}(-\frac{\mu_2}{\sqrt{2}\sigma_2})}{1 - \mathrm{erf}(-\frac{\mu_1}{\sqrt{2}\sigma_1})} \right),$$

where $\mathrm{erf}(\cdot)$ is the Gauss error function.

Let $\mu_1 = 0$ and $\sigma_1$ limit to zero. We can obtain that

$$\lim_{\sigma_1 \to 0} \mathrm{KL}(\mathcal{TN}_1(0, \sigma_1) \| \mathcal{TN}_2(\mu_2, \sigma_2)) = -\log \sigma_2 + \frac{\mu_2^2}{2\sigma_2^2} + \log(1 - \mathrm{erf}(-\frac{\mu_2}{\sqrt{2}\sigma_2})).$$

## F    THE ALGORITHM OF LOGICAL TRAINING

In practice, we set a lower bound (0.01) for the variance $\delta^2$ considering the numerical stability.

## G    OPTIMALITY OF ALGORITHM 1

### G.1    CONVERGENCE OF DUAL VARIABLES

We first discuss the optimality of dual variables $(\tau_\wedge^*, \tau_\vee^*)$. Since $\mu$ is convex w.r.t. to $\tau_\wedge$ and $\tau_\vee$, and $L(\mathbf{w}, \boldsymbol{\delta}; \tau_\wedge, \tau_\vee)$ is strictly increasing and convex w.r.t. $\mu_i$ on $[0, +\infty)$, we can derive that

**Algorithm 1** Logical Training Procedure

---

**Initialize:** $\mathbf{w}^0$ randomly; $\boldsymbol{\tau}_\wedge^0$ and $\boldsymbol{\tau}_\vee^0$ uniformly; $\boldsymbol{\delta}^0 = 1$.
**for** $t = 0, 1, \ldots,$ **do**
    Draw a collection of i.i.d. data samples $\{(\mathbf{x}_i, \mathbf{y}_i)\}_{i=1}^N$.
    $\mathbf{w}^{t+1} \leftarrow \mathbf{w}^t - \eta_\mathbf{w} \cdot \nabla_\mathbf{w} L(\mathbf{w}, \boldsymbol{\delta}; \boldsymbol{\tau}_\wedge, \boldsymbol{\tau}_\vee)$.
    $\boldsymbol{\delta}^{t+1} \leftarrow \sqrt{(\sum_{i=1}^N \boldsymbol{\mu}_i^t)/N}$.
    $\boldsymbol{\tau}_\wedge^{t+1} \leftarrow \boldsymbol{\tau}_\wedge^t + \eta_\wedge \cdot \nabla_{\boldsymbol{\tau}_\wedge} L(\mathbf{w}, \boldsymbol{\delta}; \boldsymbol{\tau}_\wedge, \boldsymbol{\tau}_\vee)$.
    $\boldsymbol{\tau}_\vee^{t+1} \leftarrow \boldsymbol{\tau}_\vee^t - \eta_\vee \cdot \nabla_{\boldsymbol{\tau}_\vee} L(\mathbf{w}, \boldsymbol{\delta}; \boldsymbol{\tau}_\wedge, \boldsymbol{\tau}_\vee)$.
**end for**

---

$L(\mathbf{w}, \boldsymbol{\delta}; \boldsymbol{\tau}_\wedge, \boldsymbol{\tau}_\vee)$ is convex w.r.t. $\boldsymbol{\tau}_\wedge$ and $\boldsymbol{\tau}_\vee$, via the convexity of composite functions (Boyd et al., 2004, Sec. 3.2). Hence, $\min_{\boldsymbol{\tau}_\vee} \max_{\boldsymbol{\tau}_\wedge} L(\mathbf{w}, \boldsymbol{\delta}; \boldsymbol{\tau}_\wedge, \boldsymbol{\tau}_\vee)$ is in fact a convex-concave optimization, and the PL assumption is satisfied (Yang et al., 2021), thus the GDA algorithm with suitable step size can achieve a global minimax point (i.e., saddle point) in $O(\varepsilon^{-2})$ iterations (Nedić & Ozdaglar, 2009; Adolphs, 2018). Some more properties of $(\boldsymbol{\tau}_\wedge^*, \boldsymbol{\tau}_\vee^*)$ are also detailed in Proposition 1, 2, and Theorem 1.

Next, we confirm that, the PL condition holds when the logical constraints are not sufficiently satisfied. Since $L(\mathbf{w}, \boldsymbol{\delta}; \boldsymbol{\tau}_\wedge, \boldsymbol{\tau}_\vee)$ is strictly increasing and convex w.r.t. $\boldsymbol{\mu}_i$ on $[0, +\infty)$, we instead analyze the cost function, i.e., $\boldsymbol{\mu}_i = S_\alpha(\mathbf{v})$.

**Proposition 4.** *Given logical constraint* $\alpha$, *assume its corresponding cost function is* $S_\alpha(\mathbf{v})$, *and* $\max_{\boldsymbol{\tau}_\wedge} \min_{\boldsymbol{\tau}_\vee} S_\alpha(\mathbf{v}) \geq \kappa$ *with constant* $\kappa > 0$. *Then, the PL property for any* $\boldsymbol{\tau}_\wedge$, *i.e.,*

$$\|\nabla_{\boldsymbol{\tau}_\wedge} S_\alpha(\boldsymbol{v}; \boldsymbol{\tau}_\wedge, \boldsymbol{\tau}_\vee)\|^2 \geq \kappa[\max_{\boldsymbol{\tau}_\wedge} S_\alpha(\boldsymbol{v}; \boldsymbol{\tau}_\wedge, \boldsymbol{\tau}_\vee) - S_\alpha(\boldsymbol{v}; \boldsymbol{\tau}_\wedge, \boldsymbol{\tau}_\vee)]$$

*holds for any* $\boldsymbol{\tau}_\wedge$.

*Proof.* Let $t_{ij} = \max(v_{ij} - c_{ij}, 0)$, we have

$$\|\nabla_{\boldsymbol{\tau}_\wedge} S_\alpha(\boldsymbol{v}; \boldsymbol{\tau}_\wedge, \boldsymbol{\tau}_\vee)\|^2 = \sum_{i \in \mathcal{I}} (\sum_{j \in \mathcal{J}} \nu_{ij} t_{ij})^2,$$

and

$$\max_{\boldsymbol{\tau}_\wedge} S_\alpha(\boldsymbol{v}; \boldsymbol{\tau}_\wedge, \boldsymbol{\tau}_\vee) = \max_{i \in \mathcal{I}} (\sum_{j \in \mathcal{J}} \nu_{ij} t_{ij}).$$

Since

$$\max_{i \in \mathcal{I}} \sum_{j \in \mathcal{J}} \nu_{ij} t_{ij} \geq \max_{i \in \mathcal{I}} \min_{j \in \mathcal{J}} t_{ij} \geq \kappa,$$

we can obtain that

$$\|\nabla_{\boldsymbol{\tau}_\wedge} S_\alpha(\boldsymbol{v}; \boldsymbol{\tau}_\wedge, \boldsymbol{\tau}_\vee)\|^2 \geq (\max_{i \in \mathcal{I}} \sum_{j \in \mathcal{J}} \nu_{ij} t_{ij})^2 \geq \kappa \max_{i \in \mathcal{I}} \sum_{j \in \mathcal{J}} \nu_{ij} t_{ij} = \kappa \max_{\boldsymbol{\tau}_\wedge} S_\alpha(\boldsymbol{v}; \boldsymbol{\tau}_\wedge, \boldsymbol{\tau}_\vee).$$

Now, we can complete the proof by using the non-negativity of $S_\alpha(\boldsymbol{v}; \boldsymbol{\tau}_\wedge, \boldsymbol{\tau}_\vee)$. □

### G.2 CONVERGENCE OF MODEL PARAMETERS

For the minimization of $L(\mathbf{w}, \boldsymbol{\delta}; \boldsymbol{\tau}_\wedge, \boldsymbol{\tau}_\vee)$ w.r.t. $\mathbf{w}$ and $\boldsymbol{\delta}$, it can be viewed as a competitive optimization with

$$\min_\mathbf{w} \ell_{\text{training}}(\mathbf{w}) + \ell_{\text{logic}}(\mathbf{w}, \boldsymbol{\delta}), \quad \min_{\boldsymbol{\delta}} \ell_{\text{logic}}(\mathbf{w}, \boldsymbol{\delta}),$$

where $\ell_{\text{training}}(\mathbf{w})$ and $\ell_{\text{logic}}(\mathbf{w}, \boldsymbol{\delta})$ are training loss and logical loss, i.e.,

$$\ell_{\text{training}}(\mathbf{w}) = \sum_{i=1}^N \text{KL}(p(\mathbf{y}_i \mid \mathbf{x}_i) \| p_\mathbf{w}(\mathbf{y}_i \mid \mathbf{x}_i)),$$

$$\ell_{\text{logic}}(\mathbf{w}, \boldsymbol{\delta}) = \sum_{i=1}^N \mathbb{E}_{\mathbf{y}_i \mid \mathbf{x}_i}[\text{KL}(p(\mathbf{z}_i \mid \mathbf{x}_i, \mathbf{y}_i) \| p_\mathbf{w}(\mathbf{z}_i \mid \mathbf{x}_i, \mathbf{y}_i))].$$

A direct method to solve this problem is conducting the gradient descent on $(\mathbf{w}, \boldsymbol{\delta})$ together. However, it is highly inefficient since we have to use a smaller step size to ensure the convergence (Lu et al., 2019). An alternative choice is alternating gradient descent on $\mathbf{w}$ and $\boldsymbol{\delta}$, but it may converge to a limit cycle or a saddle point (Powell, 1973). Algorithm 1 can indeed ensure $\mathbf{w}^*$ to be an approximately stationary point (i.e., the norm of its gradient is small). Follow the proof of Davis & Drusvyatskiy (2018, Theorem 2.1), we present the convergence guarantee as follows.

We start with the weakly convexity of function $\min_{\mathbf{y}} f(\cdot, \mathbf{y})$.

**Proposition 5.** *Suppose $f \colon \mathcal{X} \times \mathcal{Y} \to \mathbb{R}$ is $\ell_f$-smooth, and $\psi(\cdot) = \arg\min_{\mathbf{y}} f(\cdot, \mathbf{y})$ is continuously differentiable with Lipschitz constant $L_\psi$, then $\upsilon(\cdot) = \min_{\mathbf{y}} f(\cdot, \mathbf{y})$ is $\varrho$-weakly convex, where $\varrho = \ell_f(1 + L_\psi)$.*

*Proof.* Let $\bar{\mathbf{y}}' = \psi(\mathbf{x}')$ and $\bar{\mathbf{y}} = \psi(\mathbf{x})$. Since $f$ is $\ell_f$-smooth, we can obtain that

$$
\begin{aligned}
\upsilon(\mathbf{x}') = f(\mathbf{x}', \bar{\mathbf{y}}') &\geq f(\mathbf{x}, \bar{\mathbf{y}}) + \langle \nabla_{\mathbf{x}} f(\mathbf{x}, \bar{\mathbf{y}}), \mathbf{x}' - \mathbf{x} \rangle \\
&\quad + \langle \nabla_{\mathbf{y}} f(\mathbf{x}, \bar{\mathbf{y}}), \bar{\mathbf{y}}' - \bar{\mathbf{y}} \rangle - \frac{\ell_f}{2}(\|\mathbf{x}' - \mathbf{x}\|^2 + \|\bar{\mathbf{y}}' - \bar{\mathbf{y}}\|^2) \\
&\geq \upsilon(\mathbf{x}) + \langle \nabla_{\mathbf{x}} \upsilon(\mathbf{x}), \mathbf{x}' - \mathbf{x} \rangle - \frac{\ell_f}{2}\|\mathbf{x} - \mathbf{x}'\|^2 - \frac{\ell_f L_\psi}{2}\|\mathbf{x} - \mathbf{x}'\|^2,
\end{aligned}
\tag{12}
$$

which finishes the proof. $\qquad\square$

The $\rho$-weakly convexity of $\upsilon(\cdot)$ implies that $\upsilon(\mathbf{x}) + (\varrho/2)\|\mathbf{x}\|^2$ is a convex function of $\mathbf{x}$. Next, we introduce the Moreau envelope, which plays an important role in our proof.

**Proposition 6.** *For a given closed convex function $f$ from a Hibert space $\mathcal{H}$, the Moreau envelope of $f$ is defined by*

$$
e_{tf}(x) = \min_{y \in \mathcal{H}} \left\{ f(y) + \frac{1}{2t}\|y - x\|^2 \right\},
$$

*where $\|\cdot\|$ is the usual Euclidean norm. The minimizer of the Moreau envelope $e_{tf}(x)$ is called the proximal mapping of $f$ at $x$, and we denote it by*

$$
\mathrm{Prox}_{tf}(x) = \arg\min_{y \in \mathcal{H}} \left\{ f(y) + \frac{1}{2t}\|y - x\|^2 \right\}.
$$

*It is proved in Rockafellar (2015, Theorem 31.5) that the envelope function $e_{tf}(\cdot)$ is convex and continuously differentiable with*

$$
\nabla e_{tf}(x) = \frac{1}{t}\big(x - \mathrm{Prox}_{tf}(x)\big).
\tag{13}
$$

The following theorem bridges the Moreau envelope and the subdifferential of a weakly convex function (Jin et al., 2020, Lemma 30). For details, a small gradient $\|\nabla e_\upsilon(\mathbf{x})\|$ implies that $\mathbf{x}$ is close to a nearly stationary point $\widehat{\mathbf{x}}$ of $\upsilon(\cdot)$.

**Theorem 4.** *Assume the function $\upsilon$ is $\varrho$-weakly convex. For any $\lambda < 1/\varrho$, let $\widehat{\mathbf{x}} = \mathrm{Prox}_{\lambda\upsilon}(\mathbf{x})$. If $\|\nabla e_{\lambda\upsilon}(\mathbf{x})\| \leq \epsilon$, then*

$$
\|\widehat{\mathbf{x}} - \mathbf{x}\| \leq \lambda\epsilon, \quad and \quad \min_{\boldsymbol{g} \in \partial \upsilon(\widehat{\mathbf{x}})} \|\boldsymbol{g}\| \leq \epsilon.
$$

This theorem is an immediate result of the fact that stationary points of $\upsilon(\cdot)$ coincide with those of the smooth function $e_{\lambda\upsilon}(\cdot)$, and one can refer to Drusvyatskiy & Paquette (2019, Lemma 4.3) for more details. Now, we are ready to prove the convergence of Algorithm 1.

**Theorem 5.** *Suppose $f$ is $\ell_f$-smooth and $L_f$-Lipschitz, define $\upsilon(\cdot) = \min_{\mathbf{y}} f(\cdot, \mathbf{y})$ and $\psi(\cdot) = \arg\min_{\mathbf{y}} f(\cdot, \mathbf{y})$. The iterative updating of minimization $\min_{\mathbf{x},\mathbf{y}} f(\mathbf{x}, \mathbf{y})$ is*

$$
\begin{aligned}
\mathbf{x}_{t+1} &= \mathbf{x}_t - \eta \nabla_{\mathbf{x}} f(\mathbf{x}, \mathbf{y}), \\
\mathbf{y}_{t+1} &= \varphi(\mathbf{x}_{t+1}), \quad s.t. \quad \|\mathbf{y}_{t+1} - \psi(\mathbf{x}_{t+1})\| \leq \epsilon.
\end{aligned}
$$

*Assume that $\psi(\cdot)$ and $\varphi(\cdot)$ are Lipschitzian with modules $L_\psi$ and $L_\varphi$, and let $\varrho = \ell_f(1 + L_\psi)$ and $\rho = \ell_f(1 + 2L_\varphi^2)$. Then, with step size $\eta = \gamma/\sqrt{T+1}$, the output $\bar{\mathbf{x}}$ of $T$ iterations satisfies*

$$
\mathbb{E}[\|\nabla e_{\upsilon/2\rho}(\bar{\mathbf{x}})\|^2] \leq \frac{4\varrho^2}{\rho(\rho + \varrho)} \left( \frac{(e_{\upsilon/2\rho}(\mathbf{x}_0) - \min_{\mathbf{x}} \upsilon(\mathbf{x})) + \rho\gamma^2 L_f^2}{\gamma\sqrt{T+1}} + \rho\ell_f\epsilon \right),
$$

*when $\rho \geq \varrho$, and*

$$\mathbb{E}[\|\nabla e_{\upsilon/2\varrho}(\bar{\mathbf{x}})\|] \leq \frac{4\varrho^2}{\varrho(3\varrho - \rho)} \left( \frac{\left(e_{\upsilon/2\varrho}(\mathbf{x}_0) - \min_{\mathbf{x}} \upsilon(\mathbf{x})\right) + \varrho\gamma^2 L_f^2}{\gamma\sqrt{T+1}} + \varrho\ell_f\epsilon \right),$$

*otherwise.*

*Proof.* We have

$$
\begin{aligned}
\upsilon(\mathbf{x}') = f(\mathbf{x}', \bar{\mathbf{y}}') &\geq f(\mathbf{x}', \mathbf{y}_t) - \langle \nabla_{\mathbf{y}} f(\mathbf{x}', \bar{\mathbf{y}}'), \mathbf{y}_t - \bar{\mathbf{y}}' \rangle - \frac{\ell_f}{2}\|\mathbf{y}_t - \bar{\mathbf{y}}'\|^2 \\
&= f(\mathbf{x}', \mathbf{y}_t) - \frac{\ell_f}{2}\|\mathbf{y}_t - \bar{\mathbf{y}}'\|^2 \\
&\geq f(\mathbf{x}_t, \mathbf{y}_t) + \langle \nabla_{\mathbf{x}} f(\mathbf{x}_t, \mathbf{y}_t), \mathbf{x}' - \mathbf{x}_t \rangle - \frac{\ell_f}{2}\|\mathbf{x}' - \mathbf{x}_t\|^2 - \frac{\ell_f}{2}\|\mathbf{y}_t - \bar{\mathbf{y}}'\|^2 \\
&\geq \upsilon(\mathbf{x}_t) + \langle \nabla_{\mathbf{x}} f(\mathbf{x}_t, \mathbf{y}_t), \mathbf{x}' - \mathbf{x}_t \rangle - \frac{\ell_f}{2}\|\mathbf{x}' - \mathbf{x}_t\|^2 - \frac{\ell_f}{2}\|\mathbf{y}_t - \bar{\mathbf{y}}'\|^2.
\end{aligned}
$$

Since

$$
\begin{aligned}
\|\mathbf{y}_t - \bar{\mathbf{y}}'\|^2 = \|\mathbf{y}_t - \psi(\mathbf{x}')\|^2 &\leq 2(\|\mathbf{y}_t - \varphi(\mathbf{x}')\| + \|\varphi(\mathbf{x}') - \psi(\mathbf{x}')\|)^2 \\
&\leq 2(\|\mathbf{y}_t - \varphi(\mathbf{x}')\|^2 + \epsilon) = 2(\|\varphi(\mathbf{x}_t) - \varphi(\mathbf{x}')\|^2 + \epsilon) \leq 2(L_\varphi^2\|\mathbf{x}_t - \mathbf{x}'\|^2 + \epsilon),
\end{aligned}
$$

where the second inequality is derived by using the Cauchy-Schwarz inequality. Hence, we have

$$\upsilon(\mathbf{x}') \geq \upsilon(\mathbf{x}_t) - \ell_f\epsilon + \langle \nabla_{\mathbf{x}} f(\mathbf{x}_t, \mathbf{y}_t), \mathbf{x}' - \mathbf{x}_t \rangle - \left(\frac{\ell_f + 2\ell_f L_\varphi^2}{2}\right)\|\mathbf{x}' - \mathbf{x}_t\|^2. \tag{14}$$

Let $\rho = \ell_f + 2\ell_f L_\varphi^2$. Next, we discuss two cases of $\rho \geq \varrho$ and $\rho \leq \varrho$, respectively.

- If $\rho \geq \varrho$, then let $\widehat{\mathbf{x}}_t = \text{Prox}_{\upsilon/2\rho}(\mathbf{x}_t) = \arg\min_{\mathbf{x}} \upsilon(\mathbf{x}) + \rho\|\mathbf{x} - \mathbf{x}_t\|^2$. We can obtain that

$$
\begin{aligned}
e_{\upsilon/2\rho}(\mathbf{x}_{t+1}) &= \min_{\mathbf{x}} \left\{ \upsilon(\mathbf{x}) + \rho\|\mathbf{x} - \mathbf{x}_{t+1}\|^2 \right\} \\
&\leq \upsilon(\widehat{\mathbf{x}}_t) + \rho\|\mathbf{x}_{t+1} - \widehat{\mathbf{x}}_t\|^2 \\
&= \upsilon(\widehat{\mathbf{x}}_t) + \rho\|\mathbf{x}_t - \eta\nabla_{\mathbf{x}} f(\mathbf{x}_t, \mathbf{y}_t) - \widehat{\mathbf{x}}_t\|^2 \\
&= \upsilon(\widehat{\mathbf{x}}_t) + \rho\|\mathbf{x}_t - \widehat{\mathbf{x}}_t\|^2 + 2\eta\rho \langle \nabla_{\mathbf{x}} f(\mathbf{x}_t, \mathbf{y}_t), \widehat{\mathbf{x}}_t - \mathbf{x}_t \rangle + \eta^2\rho\|\nabla_{\mathbf{x}} f(\mathbf{x}_t, \mathbf{y}_t)\|^2 \\
&= e_{\upsilon/2\rho}(\mathbf{x}_t) + 2\eta\rho \langle \nabla_{\mathbf{x}} f(\mathbf{x}_t, \mathbf{y}_t), \widehat{\mathbf{x}}_t - \mathbf{x}_t \rangle + \eta^2\rho\|\nabla_{\mathbf{x}} f(\mathbf{x}_t, \mathbf{y}_t)\|^2 \\
&\leq e_{\upsilon/2\rho}(\mathbf{x}_t) + 2\eta\rho \left( \upsilon(\widehat{\mathbf{x}}_t) - \upsilon(\mathbf{x}_t) + \ell_f\epsilon + \frac{\rho}{2}\|\mathbf{x}_t - \widehat{\mathbf{x}}_t\|^2 \right) + \eta^2\rho L_f^2,
\end{aligned}
$$

and the last inequality is derived by Eq. equation 14. Taking a telescopic sum over $t$, we have

$$e_{\upsilon/2\rho}(\mathbf{x}_T) \leq e_{\upsilon/2\rho}(\mathbf{x}_0) + 2\eta\rho \sum_{t=0}^{T} \left( \upsilon(\widehat{\mathbf{x}}_t) - \upsilon(\mathbf{x}_t) + \ell_f\epsilon + \frac{\rho}{2}\|\mathbf{x}_t - \widehat{\mathbf{x}}_t\|^2 \right) + \eta^2\rho L_f^2 T.$$

Rearranging this, we obtain that

$$\frac{1}{T+1} \sum_{t=0}^{T} \left( \upsilon(\mathbf{x}_t) - \upsilon(\widehat{\mathbf{x}}_t) - \frac{\rho}{2}\|\mathbf{x}_t - \widehat{\mathbf{x}}_t\|^2 \right) \leq \frac{e_{\upsilon/2\rho}(\mathbf{x}_0) - \min_{\mathbf{x}} \upsilon(\mathbf{x})}{2\eta\rho T} + \ell_f\epsilon + \frac{\eta L_f^2}{2}. \tag{15}$$

Since $\upsilon(\mathbf{x})$ is $(\varrho/2)$-weakly convex, and thus $\upsilon(\mathbf{x}) + \rho\|\mathbf{x} - \mathbf{x}_t\|^2$ is $(\varrho/2)$ strongly convex when $\rho \geq \varrho$. Therefore, we can obtain that

$$
\begin{aligned}
\upsilon(\mathbf{x}_t) &- \upsilon(\widehat{\mathbf{x}}_t) - \frac{\rho}{2}\|\mathbf{x}_t - \widehat{\mathbf{x}}_t\|^2 \\
&= \upsilon(\mathbf{x}_t) + \rho\|\mathbf{x}_t - \mathbf{x}_t\|^2 - \upsilon(\widehat{\mathbf{x}}_t) - \rho\|\widehat{\mathbf{x}}_t - \mathbf{x}_t\|^2 + \frac{\rho}{2}\|\mathbf{x}_t - \widehat{\mathbf{x}}_t\|^2 \\
&= \left( \upsilon(\mathbf{x}_t) + \rho\|\mathbf{x}_t - \mathbf{x}_t\|^2 - \min_{\mathbf{x}} \left\{ \upsilon(\mathbf{x}) + \rho\|\mathbf{x} - \mathbf{x}_t\|^2 \right\} \right) + \frac{\rho}{2}\|\mathbf{x}_t - \widehat{\mathbf{x}}_t\|^2 \\
&\geq \frac{\rho + \varrho}{2}\|\mathbf{x}_t - \widehat{\mathbf{x}}_t\|^2 = \frac{\rho + \varrho}{8\varrho^2}\|\nabla e_{\upsilon/2\varrho}(\mathbf{x}_t)\|^2,
\end{aligned}
$$

where the last equation holds by using Eq. equation 13. One can prove the result by combining this with Eq. equation 15.

- If $\rho \leq \varrho$, then let $\widehat{\mathbf{x}}_t = \text{Prox}_{\upsilon/2\varrho}(\mathbf{x}_t) = \arg\min_{\mathbf{x}} \upsilon(\mathbf{x}) + \varrho\|\mathbf{x} - \mathbf{x}_t\|^2$. We can obtain that

$$
\begin{aligned}
e_{\upsilon/2\varrho}(\mathbf{x}_{t+1}) &= \min_{\mathbf{x}} \left\{ \upsilon(\mathbf{x}) + \varrho\|\mathbf{x} - \mathbf{x}_{t+1}\|^2 \right\} \\
&\leq \upsilon(\widehat{\mathbf{x}}_t) + \varrho\|\mathbf{x}_{t+1} - \widehat{\mathbf{x}}_t\|^2 \\
&= \upsilon(\widehat{\mathbf{x}}_t) + \varrho\|\mathbf{x}_t - \eta\nabla_{\mathbf{x}}f(\mathbf{x}_t, \mathbf{y}_t) - \widehat{\mathbf{x}}_t\|^2 \\
&= \upsilon(\widehat{\mathbf{x}}_t) + \varrho\|\mathbf{x}_t - \widehat{\mathbf{x}}_t\|^2 + 2\eta\varrho\langle\nabla_{\mathbf{x}}f(\mathbf{x}_t, \mathbf{y}_t), \widehat{\mathbf{x}}_t - \mathbf{x}_t\rangle + \eta^2\varrho\|\nabla_{\mathbf{x}}f(\mathbf{x}_t, \mathbf{y}_t)\|^2 \\
&= e_{\upsilon/2\varrho}(\mathbf{x}_t) + 2\eta\varrho\langle\nabla_{\mathbf{x}}f(\mathbf{x}_t, \mathbf{y}_t), \widehat{\mathbf{x}}_t - \mathbf{x}_t\rangle + \eta^2\varrho\|\nabla_{\mathbf{x}}f(\mathbf{x}_t, \mathbf{y}_t)\|^2 \\
&\leq e_{\upsilon/2\varrho}(\mathbf{x}_t) + 2\eta\varrho\left(\upsilon(\widehat{\mathbf{x}}_t) - \upsilon(\mathbf{x}_t) + \ell_f\epsilon + \frac{\rho}{2}\|\mathbf{x}_t - \widehat{\mathbf{x}}_t\|^2\right) + \eta^2\varrho L_f^2.
\end{aligned}
$$

Taking a telescopic sum over $t$, we have

$$
e_{\upsilon/2\varrho}(\mathbf{x}_T) \leq e_{\upsilon/2\varrho}(\mathbf{x}_0) + 2\eta\varrho\sum_{t=0}^{T}\left(\upsilon(\widehat{\mathbf{x}}_t) - \upsilon(\mathbf{x}_t) + \ell_f\epsilon + \frac{\rho}{2}\|\mathbf{x}_t - \widehat{\mathbf{x}}_t\|^2\right) + \eta^2\varrho L_f^2 T.
$$

Rearranging this, we obtain that

$$
\frac{1}{T+1}\sum_{t=0}^{T}\left(\upsilon(\mathbf{x}_t) - \upsilon(\widehat{\mathbf{x}}_t) - \frac{\rho}{2}\|\mathbf{x}_t - \widehat{\mathbf{x}}_t\|^2\right) \leq \frac{e_{\upsilon/2\varrho}(\mathbf{x}_0) - \min_{\mathbf{x}} \upsilon(\mathbf{x})}{2\eta\varrho T} + \ell_f\epsilon + \frac{\eta L_f^2}{2}. \quad (16)
$$

Since $\upsilon(\mathbf{x})$ is $(\varrho/2)$-weakly convex, $\upsilon(\mathbf{x}) + \varrho\|\mathbf{x} - \mathbf{x}_t\|^2$ is $(\varrho/2)$ strongly convex. Then, we can obtain that

$$
\begin{aligned}
&\upsilon(\mathbf{x}_t) - \upsilon(\widehat{\mathbf{x}}_t) - \frac{\rho}{2}\|\mathbf{x}_t - \widehat{\mathbf{x}}_t\|^2 \\
&= \upsilon(\mathbf{x}_t) + \varrho\|\mathbf{x}_t - \mathbf{x}_t\|^2 - \upsilon(\widehat{\mathbf{x}}_t) - \varrho\|\widehat{\mathbf{x}}_t - \mathbf{x}_t\|^2 + (\varrho - \frac{\rho}{2})\|\mathbf{x}_t - \widehat{\mathbf{x}}_t\|^2 \\
&= \left(\upsilon(\mathbf{x}_t) + \varrho\|\mathbf{x}_t - \mathbf{x}_t\|^2 - \min_{\mathbf{x}}\left\{\upsilon(\mathbf{x}) + \varrho\|\mathbf{x} - \mathbf{x}_t\|^2\right\}\right) + (\varrho - \frac{\rho}{2})\|\mathbf{x}_t - \widehat{\mathbf{x}}_t\|^2 \\
&\geq (\varrho + \frac{\varrho - \rho}{2})\|\mathbf{x}_t - \widehat{\mathbf{x}}_t\|^2 = \frac{3\varrho - \rho}{8\varrho^2}\|\nabla e_{\upsilon/2\varrho}(\mathbf{x}_t)\|^2,
\end{aligned}
$$

and we finish the proof with plugging this into Eq. equation 16. $\qquad\square$

Theorem 3 can be derived by setting

$$
\begin{cases}
\Delta_0 = e_{\upsilon/2\rho}(\mathbf{x}_0) - \min_{\mathbf{x}} \upsilon(\mathbf{x}), & \kappa = \frac{4\varrho^2}{\rho(\rho+\varrho)} & \text{if} \quad \rho < \varrho, \\
\Delta_0 = e_{\upsilon/2\varrho}(\mathbf{x}_0) - \min_{\mathbf{x}} \upsilon(\mathbf{x}), & \kappa = \frac{4\varrho^2}{\varrho(3\varrho-\rho)} & \text{if} \quad \rho \geq \varrho.
\end{cases}
$$

Finally, we confirm the assumptions in our theorems. To ensure the Lipschitz continuity of $L(\mathbf{w}, \boldsymbol{\delta})$ in our problem, we practically impose an interval $[\boldsymbol{\delta}_{\text{lower}}, \boldsymbol{\delta}_{\text{upper}}]$. Furthermore, although we can obtain the closed-form expression of $\varphi(\mathbf{w})$, i.e.,

$$
\varphi(\mathbf{w}) = \frac{1}{N}\sum_{i=1}^{N}\boldsymbol{\mu}_i,
$$

it is difficult to compute the closed-form expression of $\psi(\mathbf{w}) = \arg\min_{\boldsymbol{\delta}} L(\mathbf{w}, \boldsymbol{\delta})$ in our work. However, for any $\bar{\boldsymbol{\delta}} = \psi(\mathbf{w})$, it holds that

$$
\bar{\boldsymbol{\delta}} = \frac{1}{N}\left(\sum_{i=1}^{N}\boldsymbol{\mu}_i^2 - \frac{\boldsymbol{\mu}_i \circ \bar{\boldsymbol{\delta}} \circ \phi(-\frac{\boldsymbol{\mu}_i}{\bar{\boldsymbol{\delta}}})}{1 - \Phi(-\frac{\boldsymbol{\mu}_i}{\bar{\boldsymbol{\delta}}})}\right),
$$

where $\circ$ means the Hadamard product. Since $\boldsymbol{\mu}_i \circ \phi(-\frac{\boldsymbol{\mu}_i}{\bar{\boldsymbol{\delta}}})$ is bounded, we can obtain that $\psi(\mathbf{w}) - \bar{\boldsymbol{\delta}}$ is also bounded, and thus satisfies the assumption in Theorem 5.

# H ADDITIONAL DETAILS FOR EXPERIMENTS

We implemented our approach via the PyTorch DL framework. For PD and DL2, we use the code provided by the respective authors. The experiments were conducted on a GPU server with two Intel Xeon Gold 5118 CPU@2.30GHz, 400GB RAM, and 9 GeForce RTX 2080 Ti GPUs. The server ran Ubuntu 16.04 with GNU/Linux kernel 4.4.0.

**Handwritten Digit Recognition.** For this experiment, we used the LeNet-5 architecture, set the batch size to 128, the number of epochs to 60. For the baseline, DL2, and our approach, we optimized the loss using Adam optimizer with learning rate 1e-3. For the PD method, we direct follow the hyper-parameters provided in its Github repository. For the SL method, we set the weight of constraint loss by 0.5, and optimized the loss using Adam optimizer with learning rate 5e-4 (which is used in its Github repository).

**Handwritten Formula Recognition.** For this experiment, we used the LeNet-5 architecture, set the batch size to 128, and fixed the number of epochs to 600. For the baseline, DL2, and our approach, we optimized the loss using Adam optimizer with learning rate 1e-3 and weight decay 1e-5. For the PD method, we direct follow the hyper-parameters provided in its Github repository.

**Shortest Path Distance Prediction.** We used the multilayer perceptron with $|V| \times |V|$ input neurons, three hidden layers with 1,000 neurons each, and an output layer of $|V|$ neurons. We used the first node (with the smallest index) as the source node. The input is the adjacency matrix of the graph and the output is the distance from the source node to all the other nodes. We applied ReLU activations for each hidden layer and output layer (to make the prediction non-negative). The batch size and the number of epochs were set to 128 and 300, respectively. The network was optimized using Adam optimizer with learning rate 1e-4 and weight decay 5e-4.

**CIAFR100 Image Classification.** In this task, we set batch size by 128, and the number of epochs by 3,600. We trained the baseline model by SGD algorithm with learning rate 0.1, and set the learning rate decay ratio by 0.1 for each 300 epochs. For other methods, we used Adam optimizer with learning rate 5e-4.

# I TRAINING EFFICIENCY BY INJECTING LOGICAL KNOWLEDGE

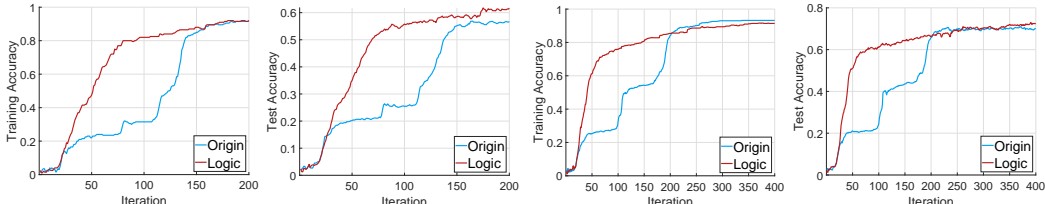

(a) The training and test curves using 2% labeled data. (b) The training and test curves using 5% labeled data.

Figure 3: The learning curves of different settings. Our algorithm with logical constraints (the red curves) significantly boosts the training efficiency compared to the plain case (the blue curves).

On HWF dataset, we also investigate whether the logical knowledge can boost the training efficiency. In details, we use the labeled data only (i.e., 2% and 5% of training data) to train the model with/without the logical constraints. Figure 3 plots of training and test accuracy with different iterations. The results illustrate the high efficiency of our logical training algorithm compared with the plain training.

# J RESULTS OF TRANSFER LEARNING EXPERIMENT

To show the robustness of the proposed approach in transfer learning, we use the STL10 dataset to evaluate a ResNet18 model trained on the CIFAR10 dataset. We set the batch size of 100, and the number of epochs by 300. For both baseline and our approach, we remove the data augmentation

operators in training process, and optimize the loss by Adam optimizer with learning rate 1e-3. The model is trained using 3,000 labeled images and 3,000 unlabeled images (only used for our approach). We design a similar logical constraint to that in the CIFAR100 experiment. For details, we define two superclasses for CIFAR10 dataset, i.e., machines (denoted by $m$) and animals (denoted by $a$), and the constraint is formulated as

$$(p_m(\mathbf{x}) = 0.0\% \vee p_m(\mathbf{x}) = 100.0\%) \wedge (p_a(\mathbf{x}) = 0.0\% \vee p_a(\mathbf{x}) = 100.0\%).$$

The results are shown in Table 5. We do observe that the logical rule is more stable compared with model accuracy. Moreover, although such weak logical constraint only slightly improve the model (class and superclass) accuracy on CIFAR10, this increment is still preserved when domain shift occurs.

Table 5: Results from CIFAR10 to STL10.

| Datasets | Class Acc. (%) | Superclass Acc. (%) | Sat. (%) |
|----------|----------------|---------------------|----------|
| Baseline | $65.8 \rightarrow 32.6$ | $92.7 \rightarrow 76.4$ | $90.1 \rightarrow 88.4$ |
| Ours | $68.1 \rightarrow 35.4$ | $93.9 \rightarrow 78.1$ | $92.6 \rightarrow 91.9$ |

## K  RESULTS OF TRAINING EPOCH RUNTIME

For the convergence rate, Theorem 3 states that our algorithm has the same convergence order with the baseline (stochastic (sub)gradient descent) algorithm. We further show the runtime of each epoch of the CIFAR100 experiment in Table 6. We can observe that, in each epoch, our method introduces little extra computational cost.

Table 6: Runtime of each training epoch.

| Runtime | VGG16 | ResNet50 | DenseNet100 |
|---------|-------|----------|-------------|
| Baseline | 30.2s | 26.4s | 26.7s |
| Ours | 32.5s | 27.0s | 28.1s |

## L  RESULTS WITH STANDARD DEVIATION ON CIFAR100

For task 4, we include a detailed results with Standard Deviation as follows.

Table 7: Accuracy of class classification on CIFAR100 dataset.

| Method | VGG16 | ResNet50 | DenseNet100 |
|--------|-------|----------|-------------|
| Baseline | 51.0($\pm$0.52) | 46.6($\pm$1.38) | 36.2($\pm$0.61) |
| PD$_1$ | 46.7($\pm$0.99) | 52.7($\pm$0.86) | 42.3($\pm$0.67) |
| PD$_2$ | 46.9($\pm$0.10) | 54.2($\pm$0.58) | 42.4($\pm$0.48) |
| DL2 | 32.6($\pm$9.66) | 45.3($\pm$1.37) | 40.8($\pm$0.50) |
| Ours | **53.4($\pm$0.43)** | **56.0($\pm$1.78)** | **51.4($\pm$0.27)** |

Table 8: The constraint satisfaction results (%) of image classification on the CIFAR100 dataset.

| Method | VGG16 | ResNet50 | DenseNet100 |
|--------|-------|----------|-------------|
| Baseline | 63.0($\pm$0.74) | 48.5($\pm$2.37) | 39.7($\pm$1.13) |
| PD$_1$ | 57.7($\pm$3.01) | 63.9($\pm$0.62) | 53.5($\pm$0.46) |
| PD$_2$ | 57.5($\pm$1.86) | 69.2($\pm$0.90) | 54.9($\pm$0.91) |
| DL2 | 31.2($\pm$17.9) | 78.4($\pm$1.32) | 62.2($\pm$0.56) |
| Ours | **81.1($\pm$0.37)** | **87.2($\pm$0.62)** | **88.2($\pm$0.27)** |

