# OpenReview forum: "Learning with Logical Constraints but without Shortcut Satisfaction"
_ICLR.cc/2023/Conference — ICLR 2023 notable top 25%_

### Official Review · Reviewer_C2ss · 2022-10-21

**Confidence:** 4
**Correctness:** 4
**Technical Novelty And Significance:** 3
**Empirical Novelty And Significance:** 3
**Recommendation:** 8

**Clarity, Quality, Novelty And Reproducibility:**

The related work section is well-detailed, but it is missing some recent results about translating logical constraints into loss functions using straight-through estimators (e.g. Yang et. al., 2022). Yet, the present approach that uses dual variables and simplices is conceptually more simple to implement and analyze.

The clarity of the paper is remarkable. In Section 2.2, I would here suggest replacing *General Formulas* with *Clausal Formulas* (or CNF formulas). Indeed, general (negation normal form) formulas are beyond the reach of this translation method. For Theorem 3, it would be nice to add a brief comment about $\kappa$, $\gamma$, and $\Delta_0$.

**Reference:**

Yang, Z., Lee, J. and Park, C.. (2022). Injecting Logical Constraints into Neural Networks via Straight-Through Estimators. Proceedings of the 39th International Conference on Machine Learning, 2022.


**Strength And Weaknesses:**

Overall, the paper is well-motivated and well-written. I found the logic-to-loss translation in Section 2 very intuitive. Based on this translation, the use of a stochastic gradient descent-ascent learning algorithm is quite natural. The convergence analysis is, however, much more difficult to establish, and as far as I could check, the proof looks correct. Finally, experimental results reported for various constrained learning tasks highlight the merits of this framework.

I did not find any real weakness in this paper. I just have some minor recommendations, detailed below.


**Summary Of The Paper:**

In the setting of constrained deep learning, a key challenge is to inject logical constraints as loss functions while preserving the semantics of their truth tables. Notably, some logical constraints such as material implications have *preferred tuples* which should be reached whenever possible. This paper aims at addressing this challenge by providing a new translation of CNF formulas into loss functions and a variational learning framework. One of the key ideas of the translation is to map simple conjunctions and disjunctions as optimization functions into corresponding simplices. This translation satisfies desirable properties such as monotonicity, and interpretability. The variational learning framework takes the form of a nonzero-sum game for which local Nash equilibria are searched using a stochastic gradient-ascent algorithm with min-oracle. Experiments on various problems corroborate the effectiveness of this approach.


**Summary Of The Review:**

This is a conceptually simple, flexible, and robust approach to neural network learning with logical constraints. In a nutshell, a good paper.

---

> ### Author Response · Authors · 2022-11-09
> **Response to Reviewer C2ss**
>
> Thanks for the comments. We have revised the paper accordingly (e.g., we have included the mentioned paper and added a detailed comment for Theorem 3 in Appendix G).

---

### Official Review · Reviewer_UtYh · 2022-10-26

**Confidence:** 3
**Correctness:** 4
**Technical Novelty And Significance:** 4
**Empirical Novelty And Significance:** 3
**Recommendation:** 8

**Clarity, Quality, Novelty And Reproducibility:**

**Clarity**: The text is very well written, although a bit dense in parts.  I am referring especially to Section 2.1.  The meaning of the dual variables is not easy to grasp, and a couple of concrete examples would help the reader wad through that section.  The rest of the paper is easier to read, also thanks to the examples in the appendix.  The formalization is crisp.

**Quality**: This is a very solid contribution:  it tackles an important (at least to my eyes) open problem, it provides a very elegant modelization and solution procedure, a clear optimization algorithm, and plenty of evidence that the proposed approach (which, by the way, should be given a **name** for ease of reference) works as intended.  The choice of experimental task and competitors is also very appropriate.

One minor complaint is that, although the authors do mention a key limitation of their proposed method in the conclusion (namely, time complexity), they do not address possible limitations of the variational setup.  In fact, it is not entirely clear to me whether "spreading out" the model's probability mass over all logical terms/clauses is always the right approach.  This would be worth discussing briefly in the conclusion.

**Originality**:  To the best of my knowledge, no prior work has tackled shortcut satisfaction under knowledge constraints.  I believe this work will have quite an impact within the SRL and NeSy communities.


**Reproducibility**:  Experimental details are given in the appendix.  It is not clear to me whether the code will be made available upon publication, though.

**Strength And Weaknesses:**

PROS
----

- Crisply written, if a bit dense in parts.
- Tackles an underappreciated but very central issue in NeSy integration and SRL.
- Provides a very clear formulation and diagnosis of the problem.
- Proposed technique is creative and non-trivial.
- Very sensible (and surprisingly intuitive) optimization algorithm.
- Contribution is carefully compared to relevant literature.
- Empirical results are very convincing.

CONS
----

- Some parts of the text are a bit dense and would benefit from in-line concrete examples.
- Does not entirely explore limits of the proposed approach.

**Summary Of The Paper:**

The paper tackles the problem of "shortcut satisfaction" in the context of neuro-symbolic integration (their results are however of interest for statistical relational learning at large).  Shortcut satisfaction, in this context, refers to models allocating most probability mass to "easy to satisfy" variable configurations.  This is a well-known and central (but sadly overlooked) problem in NeSy and SRL.  The authors propose a technique reminiscent of T-norm approaches to constraint satisfaction, which however involves injecting dual variables into the formulation of the knowledge-based loss which control what parts of the various formulas are active.  The authors also derive an alternating SGD optimization algorithm.  Experimental results in four tasks against SOTA competitors illustrate the efficacy of the proposed approach in avoiding shortcut satisfaction while retaining high accuracy and even outperforming the competitors.

**Summary Of The Review:**

Very solid and significant contribution that tackles an overlooked open problem.

---

> ### Author Response · Authors · 2022-11-09
> **Response to Reviewer UtYh**
>
> Thanks for the comments. We have revised Section 2.1 and briefly discussed the limitation of the variational framework in the conclusion as suggested. For reproducibility, we will make the code of our method publicly available upon publication. Currently, to keep anonymity, it is accessible through the link given in the paper https://figshare.com/s/9358d95545fa25823fbc .

---

> > ### Comment · Reviewer_UtYh · 2022-11-11
> > **Replu**
> >
> > Thank you, I have no further remarks.

---

### Official Review · Reviewer_LWuq · 2022-10-26

**Confidence:** 2
**Correctness:** 3
**Technical Novelty And Significance:** 4
**Empirical Novelty And Significance:** 4
**Recommendation:** 6

**Clarity, Quality, Novelty And Reproducibility:**

Here I am just giving score, see above for details

**Clarity**: 6/10
**Quality:** 8/10
**Novelty:** 9/10
**Reproducibility:** 10/10 (code is already available, I didn't try to run it though)

**Strength And Weaknesses:**

**Strength:**

1) The paper identifies an interesting problem in the current state-of-the-art models
2) The solution proposed seems very interesting

**Weaknesses:**

1) I don't know variational learning well, but I know Neurosymbolic learning quite well and I struggled to follow the paper. A bit more explanations on the variational learning framework could really help somebody with my background.
2) The theoretical part seems to be disconnected from the experimental analysis. Indeed, in the theoretical analysis the authors state that all the atoms in the formulas are of the type $v \cdot c$, but then in the experimental analysis we have constraints of the type $v_1 +v_2 \cdot c$. Suppose we simply set $v = v_1 + v_2$ in the constraint $v \cdot c$ do all the theorems still hold? Do they also hold if we have multiple constraints sharing the same variables (e.g., $v_1 + v_2 < c$ and $v_3 - v_2 > c$).
3) In the experimental analysis the authors compare only with PD and DL2. I would suggest to also add as baseline DeepProbLog [1] as it is one of the most well known methods for including the constraints.
4) In Table 2 the authors report the results with 2% and 5% of the data. This is a very little amount. Could the authors produce the results also for 10% 20% and 50%?
5) The authors are missing some related works which they should discuss. In particular: [2,3,4,5,6].
6) Are the experiments run for only one seed? Ideally we should have 5 runs and we should have average and std. (I am asking because from what I know variational learning based methods tend to be quite unstable, so it would be useful to asses the stability of the proposed model)


**References:**

[1] Robin Manhaeve, Sebastijan Dumancic, Angelika Kimmig, Thomas Demeester, and Luc De Raedt. DeepProbLog: Neural probabilistic logic programming. In Proc. of NeurIPS, 2018.

[2] Samy Badreddine, Artur d’Avila Garcez, Luciano Serafini, and Michael Spranger. Logic tensor networks. Artif. Intell., 303, 2022.

[3] Eleonora Giunchiglia and Thomas Lukasiewicz. Multi-label classification neural networks with hard logical constraints. JAIR, 72, 2021.

[4] Tao Li and Vivek Srikumar. Augmenting neural networks with first-order logic. In Proc. of ACL, 2019.

[5] Zhun Yang, Adam Ishay, and Joohyung Lee. NeurASP: Embracing neural networks into answer set programming. In Proc. of IJCAI, 2020.

[6] Ahmed, Kareem, Stefano Teso, Kai-Wei Chang, Guy Van den Broeck, and Antonio Vergari. Semantic Probabilistic Layers for Neuro-Symbolic Learning. arXiv preprint arXiv:2206.00426, 2022.

**Summary Of The Paper:**

The paper proposes a new way of integrating logical constraints in the training of deep neural networks. In particular, the authors try to solve the problem that the current models often learn to satisfy the constraints by learning the *obvious solution* (e.g., given $A \to B$ the models learn to set $A = \bot$). The logical constraints taken in considerations by the authors are written in CNF and the atoms in the formula are $v \cdot c$, where:
- $v$ is a variable
- $\cdot \in \{ \le, \ge, <, >, =, \not = \}$
- $c$ is a constant.
In order to avoid the obvious solution the authors propose to associate a a dual variable to each logical connective.


**Summary Of The Review:**

TL;DR: The paper seems novel and interesting, it needs to improve on clarity

---

> ### Author Response · Authors · 2022-11-09
> **Response to Reviewer LWuq**
>
> Thanks for the comments. Detailed responses to the raised issues:
> 1. Variational learning: we have revised the sentences and added a reference in the revision.
> 2. Theoretical analysis: the theorems still hold when the formula involves multiple variables or multiple overlapped constraints. Specifically, our theorems are based on the state of $v$ (i.e., the satisfaction degree from $v$'s concrete instantiation to the logical constraint $v \cdot c$), be it $v = v_1 + v_2$ or others. Likewise, it can be extended to the case of multiple constraints, leading to a vector-formed state $v=[v_1+v_2; v_3-v_2]$ (details can be seen in the proof in Appendix B). We will further clarify this in the revision.
> 3. DeepProbLog: we compared Deepproblog (DPL) and semantic loss (SL) in the first experiment, and observed that these two approaches also suffer from the shortcut satisfaction issue. For the other three tasks (especially for shortest distance prediction and image classification on CIFAR100), both DPL and SL are not efficient enough (as explained in the first paragraph of Section 4) to produce sensible results.
> 4. Training data size: first, let's clarify that learning with logical constraints is especially effective when there are insufficient training data. The specific task in Table 2 is a relatively easy task. As a result, using more than 10% data (i.e., 1000 examples) without adding the logical loss would have already resulted in nearly 100% accuracy and constraint satisfaction. In other words, with more data collected in Table 2, the network can gradually learn the logical constraint from the data. (We would be willing to produce this result should the reviewer be still interested.) It is also worth noting that learning with logical constraints is still effective in more complex tasks. For example, Task 3 is under a fully supervised setting where all training data are labeled. In this task, our method can effectively regulate the neural network, and ensure its generalization performance.
> 5. Related work: we have included the suggested references in the revision.
> 6. Random seed: we have repeated each experiment 5 times (with random seeds 0-4) and reported the average (cf, "each reported experimental result is derived by computing the average of five repeats"; the first paragraph of Section 4). We have also added the std results for Task 4 in Appendix I.

---

> > ### Comment · Reviewer_LWuq · 2022-12-09
> > **Thank you for your replies**
> >
> > First of all, sorry for the delayed reply, and, secondly, thanks for having updated the paper and for the replies. Also, given more time, and the couple of changes that has been done to the paper, I could follow the variational inference part.
> >
> > As a side note, I see that you have added the citation [1] instead of [2] (as I suggested). Just as a reference [1] is a survey on how logical constraints can be introduced into deep learning methods, while [2] is a method to incorporate constraints into the loss and topology of neural networks. I thought it might have been particularly interesting for your work because by design it does not suffer of this shortcut problem, as it incorporates constraints of the type $A \wedge \neg B \to C$ and it actively teaches the neural neural network how to exploit the prediction $(A \wedge \neg C)$ to reach a conclusion on $C$. I thus thought a more in-depth analysis on which sota models suffer of these shortcuts problems and which don't would have been a good addition to the paper.
> >
> > Also, by re-reading the paper I noticed that the authors write "Although they introduce a variable to choose which constraint should be satisfied, the variable is categorical and thus limited to mutually exclusive disjunction." when speaking about [3]. This phrase is misleading as the assumption in [3] is that the constraints are expressed as a DNF formula, and thus satisfying one constraint is equivalent to satisfy them all. Going back to the point above, [3] also does not suffer from the shortcut satisfaction problem, and thus it would be a nice addition to the discussion.
> >
> > [1] Eleonora Giunchiglia, Mihaela Catalina Stoian, and Thomas Lukasiewicz. Deep learning with logical constraints. arXiv preprint arXiv:2205.00523, 2022
> >
> > [2] Eleonora Giunchiglia and Thomas Lukasiewicz. Multi-label classification neural networks with hard logical constraints. JAIR, 72, 2021.
> >
> > [3] Nicholas Hoernle, Rafael Michael Karampatsis, Vaishak Belle, and Kobi Gal. Multiplexnet: Towards fully satisfied logical constraints in neural networks. arXiv preprint arXiv:2111.01564, 2021.

---

> > > ### Author Response · Authors · 2022-12-12
> > > **Re-Response**
> > >
> > > Thanks for the reply.
> > >
> > > - We find [2] quite interesting, and will add it as suggested. We also agree that a thorough analysis of which sota models, particularly those incorporate constraints into the topology of neural networks, suffer from the shortcut problem is needed. We will conduct a more comprehensive empirical study, as well as a theoretical analysis in future work.
> > >
> > > - As to [3], we will revise the sentences for clarification. However, we conduct an initial experiment, and find that the MultiplexNet in [3] also suffers from the shortcut problem. Elaborately, in Task 1, the MultiplexNet achieves $\neg P$-Sat. =90.28% , $Q$-Sat.=15.38%, and Acc.=88.35%, which is still far from satisfactory.
> > >
> > >     The possible reasons are as follows.
> > >     - MultiplexNet introduces a categorical distribution of ${\small  P:=(f(R(\mathbf{x})) \neq 9) \vee Q:=(f(\mathbf{x})=6)}$ to indicate which term should be satisfied. However, the categorical distribution encourages a one-hot result, and thus the model will tend to satisfy the easier term $P$ for any input.
> > >     - Moreover, the marginalization used in MultiplexNet is not applicable. In this task, we want $P=T$ for images of 6 and $P=F$ for other images, but the marginalization introduces unnecessary correlation among inputs, inducing the model to hold $P = F$ for all images.
> > >
> > >     In contrast, our dual variables are respective for each input, and we do not enforce them to be one-hot, which can efficiently avoid the shortcut satisfaction.

---

### Official Review · Reviewer_X3bi · 2022-10-27

**Confidence:** 3
**Clarity, Quality, Novelty And Reproducibility:** The paper is clearly state and the co…
**Correctness:** 3
**Technical Novelty And Significance:** 3
**Empirical Novelty And Significance:** 3
**Recommendation:** 6

**Strength And Weaknesses:**

The work successfully addressed the shortcut satisfaction issue and proposed a variational framework where the logical constraint is expressed as a distributional loss that is compatible with the model’s original training loss. However the design lacks theoretical support.

**Summary Of The Paper:**

 In this paper, a new framework for learning with logical constraints is proposed. The shortcut satisfaction issue is addressed by introducing dual variables for logical connectives, encoding how the constraint is satisfied. A variational framework is proposed where the logical constraint is expressed as a distributional loss that is compatible with the model’s original training loss. Theoretical analysis shows that the proposed approach bears some nice properties, and experimental evaluations demonstrate its superior performance in both model generalizability and constraint satisfaction.

**Summary Of The Review:**

 In this paper, a new framework for learning with logical constraints is proposed. The shortcut satisfaction issue is addressed by introducing dual variables for logical connectives, encoding how the constraint is satisfied. A variational framework is proposed where the logical constraint is expressed as a distributional loss that is compatible with the model’s original training loss. Theoretical analysis shows that the proposed approach bears some nice properties, and experimental evaluations demonstrate its superior performance in both model generalizability and constraint satisfaction. The paper is clearly state and the content is well organized. The work successfully addressed the shortcut satisfaction issue and proposed a variational framework where the logical constraint is expressed as a distributional loss that is compatible with the model’s original training loss.

---

> ### Author Response · Authors · 2022-11-09
> **Response to Reviewer X3bi**
>
> Thanks for the comments. We are not quite sure about what is expected by the comment "the design lacks theoretical support". We would be grateful if the reviewer could elaborate a little bit.
>
> The theoretical soundness of our learning framework is guaranteed by the three theorems in the submission. Theorem 1 (consistency between optimal solution and logical entailment) and Theorem 2 (monotonicity of logic translation) indicate that the information provided by the logical constraints is well preserved, and Theorem 3 (convergence of the learning process) ensures the feasibility and effectiveness of the approach.
>
> We suspect that the concern is that these theorems do not explicitly mention shortcut satisfaction avoidance. If this were the case, we hope the reviewer can be reassured by noticing that the introduced dual variables provide the necessary flexibility to allow different truth assignments for different inputs, which gives a way to "smooth" the truth assignment. Via training, they are expected to attain desired truth assignment. What Theorem 1 and 2 ensure is that the desired truth assignment will be optimum, and Theorem 3 guarantees the robustness of the learning process despite the additional degree of freedom of dual variables.

---

### Decision · Program_Chairs · 2023-01-20

**Decision:**

Accept: notable-top-25%

**Justification For Why Not Higher Score:**

It is a very strong paper, but perhaps some of the related work discussion and comparison could be stronger for it to become an oral. There are many neuro-symbolic baselines.

**Justification For Why Not Lower Score:**

Best in my batch.

**Metareview: Summary, Strengths And Weaknesses:**

This paper proposes a new neuro-symbolic technique that pushes the state of the art in terms of supporting inequality constraints (i.e., relational constraints) and avoiding trivial ways of satisfying implication bodies. Presentation is clear, with clear contributions and empirical results. Obvious accept.

**Note From Pc:**

if the above contains the word "oral" or "spotlight" please see: "oral" presentation means -> notable-top-5% and "spotlight" means -> notable-top-25%. As stated in our emails, we are disassociating presentation type from AC recommendations